# WAVESCALE NEURAL AUDIO CODEC: BIDIRECTIONAL MULTISCALE RESIDUAL QUANTIZATION FOR HIGH-FIDELITY AUDIO COMPRESSION

## ABSTRACT

Modern AI systems need audio representations that are efficient in bandwidth and friendly to models. Neural codecs learn discrete token streams optimized for perceptual and task goals, unifying compression with generation, editing, retrieval and multimodal reasoning. Neural compression with residual vector quantization (RVQ) achieves low bitrates at high quality by encoding audio as discrete latents. Recent multiscale RVQ variants (e.g., SAT, SNAC) distribute quantization across multiple temporal scales to reduce token rate and computational cost; however, a purely upscale hierarchy assigns coarse (low-rate, slowly varying) structure to early stages where typically low-frequency components are assigned and fine (high-rate, rapidly varying) detail to later stages where typically high-frequency components are assigned. This works well for speech but often fails for music and environmental audio: in music, early stages can carry fine detail, whereas in environmental audio, periodicity is weak. We introduce the Wavescale Neural Audio Codec (WNAC), which replaces the pure upscale flow with a downscale then upscale path. By inserting fine-to-coarse stages before coarse-to-fine, WNAC preserves early low frequency information. We also add a scale-aware waveloss that aligns quantized outputs at the same temporal resolution across stages, improving reconstruction sharpness and stability. Experiments show higher accuracy and efficiency across speech, music, environment and a mixed general set, outperforming single-scale DAC while keeping the speed benefits of multiscale RVQ.

## 1 INTRODUCTION

Neural audio compression has recently emerged as a powerful alternative to traditional codecs such as MP3 and AAC, offering superior performance by learning compact, task-specific representations in a fully end-to-end manner (Zeghidour et al., 2021; Défossez et al., 2023). These models encode raw audio waveforms into sequences of discrete latent variables, enabling high-quality reconstruction at lower bitrates.

A key strength of this approach is its compatibility with generative models. By representing audio as discrete tokens, neural codecs bridge the gap between compression and generation, supporting tasks such as speech synthesis, music generation, and audio translation (van den Oord et al., 2017; Kreuk et al., 2022; Jiang et al., 2025). However, existing single-scale tokenizers often require many tokens per second to maintain fidelity, resulting in high computational cost and reduced generalization (Lee et al., 2024).

To mitigate this, multiscale RVQ-VAE models apply residual quantization across multiple temporal scales: early stages model slowly varying *coarse* structure, whereas later stages capture rapidly varying *fine* detail (Tian et al., 2024; Siuzdak et al., 2024). Tian et al. (2024); Qiu et al. (2024) extend multiscale RVQ to autoregressive modeling via next-scale prediction, forecasting the next *scale* rather than the next *token*; although next-token prediction is the standard AR paradigm, next-scale prediction delivers comparable performance at substantially lower computational cost.

Despite their effectiveness, these models rely on a bottom-up (coarse-to-fine) strategy that assumes low-frequency components are generally coarse in content. However, in music or sound effects,

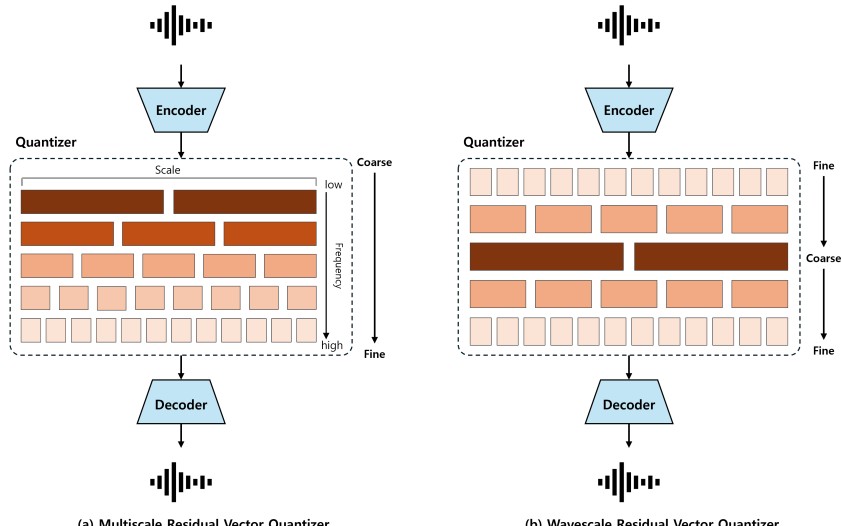

(a) Multiscale Residual Vector Quantizer  (b) Wavescale Residual Vector Quantizer

Figure 1: (a) Conventional multiscale RVQ and (b) the proposed Wavescale RVQ. RVQ greedily selects the code that maximally reduces the residual energy, early stages typically absorb low-frequency content, leaving high-frequency detail to later stages. Wavescale begins at the fine scale, then downsamples and finally refines by upsampling, preserving high-resolution cues while following the same low-to-high frequency allocation across stages.

low frequencies often encode meaningful harmonic or rhythmic content. In environmental audio where periodicity is sparse, the coarse to fine hierarchy becomes ineffective, leading to early stage information loss and reduced robustness in non-speech domains (Zheng et al., 2024). Empirical domain analyses support this claim; see Appendix B.4 for details.

To address this, we propose the WNAC, a multiscale RVQ-VAE that modified the traditional flow. By adopting a downscale-upscale configuration, our model first encodes at high resolution and progressively downsamples, preserving critical features early on. This structure improves both computational efficiency and reconstruction fidelity across four domains: speech, music, environment, and a general set that mixes all three. We also introduce a scale-aware loss (waveloss), which enforces consistency across codebooks (learned sets of prototype vectors used for quantization) operating at the same resolution, improving reconstruction sharpness and stability by minimizing the mean squared error between the quantized outputs of each pair of stages at the same scale in the wavescale RVQ.

The comparison of tranditional multiscale RVQ and wavescale RVQ is illustrated at Figure 1.

Our main contributions are as follows:

- We propose the Wavescale Neural Audio Codec, a novel multiscale residual quantization framework that departs from the conventional coarse to fine structure by introducing a fine-to-coarse downscaling followed by upscaling, enabling improved compression fidelity and reconstruction quality.
- We introduce a scale-aware loss that enforces consistency across codebooks operating at the same resolution but different stages, enhancing cross-scale information alignment.
- We evaluate our architecture across domains and analyze scale-wise latent behavior and periodicity sensitivity as well as ablations. The method delivers superior reconstruction accuracy, codebook efficiency and overall robustness.

We provide code and model weights as open-source at `https://anonymous.4open.science/r/WNAC`[1]. The reconstructed audio samples used in our experiments are provided as part of the supplementary material.

---

[1]The source code repository has been temporarily anonymized for peer review.

## 2 RELATED WORK

### 2.1 AUDIO COMPRESSION WITH RVQGAN

Vector Quantized Variational Autoencoder (VQ-VAE) (van den Oord et al., 2017) is a foundational method for learning discrete representations by mapping continuous latent variables to entries in a learnable codebook. A codebook is a finite set of embedding vectors, each representing a prototype in the latent space. During quantization, each latent vector is replaced with the nearest codebook entry, effectively discretizing the representation. This process enables compact and expressive encoding, which is well suited for tasks like audio compression and generation. Residual VQ-VAE (RVQ-VAE) (Lee et al., 2022; Zheng et al., 2024) improves on this by introducing multi-stage quantization, where each stage encodes the residual from the previous step, enabling finer detail preservation and better codebook usage. Encodec (Défossez et al., 2023) applies this strategy within a fully convolutional encoder and decoder architecture, providing flexible bitrate control and high quality reconstruction.

Descript Audio Codec (DAC) (Kumar et al., 2023) builds on RVQ-VAE using adversarial training (Goodfellow et al., 2020), forming RVQGAN. It introduces multi-scale STFT discriminators (Guo et al., 2022), Mel-spectrogram losses, and periodic activations like Snake to better model time-frequency structure in audio. These methods have advanced neural audio codecs by combining residual quantization with perceptual objectives. However, these single-scale RVQGANs operate at a same temporal resolution at every quantization stage, which limits their ability to efficiently model both low and high frequency content, often leading to redundant token usage.

### 2.2 MULTISCALE RVQGAN

Tian et al. (2024) introduced multiscale quantization to RVQGAN, applying scale-dependent interpolation at each RVQ step across the scale hierarchy and coupling it with next-scale prediction for autoregressive modeling. Unlike traditional next-token prediction, which autoregresses over the entire token sequence, next-scale prediction autoregresses only over the resolution stages; inference therefore scales with the number of scales $S$ rather than the total number of tokens $L$ (i.e., $O(S)$ vs. $O(L)$), yielding substantially lower latency. Qiu et al. (2024) proposed the Scale-level Audio Tokenizer (SAT) for audio, integrating SEANet (Zeghidour et al., 2021) and a phi kernel for improved fidelity. Siuzdak et al. (2024) further enhanced multiscale RVQ using downscaling pools, noise blocks, depthwise convolutions (Howard et al., 2017), and windowed attention (Beltagy et al., 2020), improving model robustness and reconstruction quality.

A limitation of these models lies in their bottom-up residual computation, which begins at the coarsest resolution. This assumes low-frequency components are semantically sparse, but in domains like music, low-frequency signals can carry rich harmonic or rhythmic content (Zheng et al., 2024; Lanzendörfer et al., 2024). As a result, early-stage quantization may discard critical information, limiting performance on complex or unstructured signals.

To address this limitation, we propose a **wavescale residual quantization** framework that departs from the conventional bottom-up structure. Instead of beginning quantization at the coarsest level, our model starts at the highest resolution and progressively downsamples through lower-resolution quantizers. The resulting multiscale latents are then refined through an upsampling path, allowing early preservation of fine detail and late-stage integration of semantic structure. We further introduce a cross-resolution consistency loss ($L_u$) to align latent representations across scales and enhance reconstruction quality.

## 3 METHOD

### 3.1 PRELIMINARIES

#### 3.1.1 MULTISCALE QUANTIZATION

We build on the *single-scale* RVQGAN of Kumar et al. (2023); our codebase and baseline implementation are directly derived from their setup. However, in this subsection, we formalize *multiscale*

residual vector quantization extension, which operates across a sequence of different temporal resolutions $\{T_i\}_{i=0}^{n-1}$ rather than a single scale. $T'$ is the temporal length of encoded latent vector $z_0 = \text{Encoder}(x)$. Quantization proceeds through $n$ residual stages. At stage $i$, the quantized output $q_i$ and residual $z_{i+1}$ are:

$$q_i = W_{out}S_{T'}^{T_i}(e_k), \quad k = \arg\min_j ||l_2(S_{T_i}^{T'}(W_{in}z_i)) - l_2(e_j)||_2$$

$$z_{i+1} = z_i - q_i$$

Here, $S_{T'}^{T_i}(\cdot)$ denotes the interpolation operator that adjusts the temporal resolution from $T'$ to $T_i$, and $l_2(\cdot)$ is the L2 normalization. We use $\texttt{area}$ interpolation for downscaling and $\texttt{linear}$ interpolation for upscaling, reflecting the different objectives of energy-preserving resampling and smooth reconstruction. The projection matrix $W_{\text{in}}$ maps the encoder output to an intermediate latent space, and $W_{\text{out}}$ transforms the selected codebook vector $e_k$ into the final quantized representation. Each $q_i$ captures residuals at a specific temporal scale, and the compressed codes correspond to the selected entries $e_k$. The final reconstruction is given by:

$$\hat{x} = \text{Decoder}\left(\sum_{i=0}^{n-1} q_i\right)$$

### 3.1.2 LOSS FUNCTION

Following RVQGAN, we combine reconstruction, perceptual, adversarial, and quantization losses. Codebook and Commitment Losses from standard VQ-VAE losses are defined as:

$$L_{cb} = \frac{1}{n}\sum_{i=0}^{n-1} ||\text{sg}(q_i) - z_i||_2^2, \quad L_{cm} = \frac{1}{n}\sum_{i=0}^{n-1} ||q_i - \text{sg}(z_i)||_2^2$$

where $\text{sg}(\cdot)$ denotes the stop-gradient that prevents the back-propagation of gradients through $z_i$ to separate the encoder and codebook updates. To improve the fidelity in the time and frequency domains, Waveform and Frequency Losses are defined as:

$$L_w = ||x - \hat{x}||_1, \quad L_f = \sum_i \left(||M_i(x) - M_i(\hat{x})||_1 + ||M_i(x) - M_i(\hat{x})||_2^2\right)$$

with $M_i(\cdot)$ as multiscale mel-spectrograms. Finally, adversarial loss is defined by setup of DAC (Kumar et al., 2023), with $L_g$ for generator loss and $L_d$ for discriminator feedback using multi-scale STFT. The total loss is:

$$L = \lambda_w L_w + \lambda_f L_f + \lambda_g L_g + \lambda_d L_d + \lambda_{cb} L_{cb} + \lambda_{cm} L_{cm}$$

## 3.2 WAVESCALE RESIDUAL VECTOR QUANTIZATION

Traditional multiscale RVQ-VAE architectures typically begin the quantization process at the lowest temporal resolution, progressively adding higher resolution residuals. While effective for certain types of audio signals, this strategy suffers from a critical drawback: starting from such a coarse scale can lead to significant information loss, particularly in the low frequency components of structured audio such as speech and music.

To overcome this limitation, we propose the Wavescale structure, a novel quantization framework that modifies the conventional quantization order. As illustrated in Figure 2, the encoding process begins at the highest resolution, allowing the model to immediately capture fine-grained details. The signal is then progressively downsampled, with each stage quantizing the residual between the current representation and its lower-resolution approximation. This structure enables richer initial encoding and more effective use of codebook capacity across layers. Finally, the signal is upsampled back to the original resolution, with each higher stage refining the previous coarse prediction.

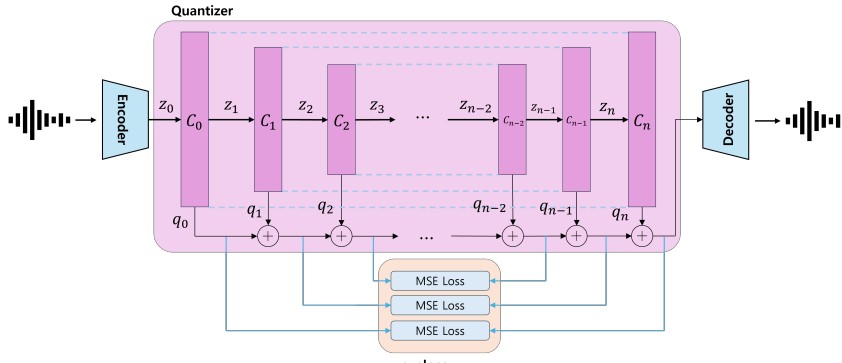

Figure 2: Our proposed **Wavescale RVQ-VAE**. Wavescale begins quantization at the finest scale, progressively downsamples, and refines via upsampling. Each quantization step consists of a codebook $C_n$, the residual input $z_n$, and the quantization result (codes) $q_n$. Codebooks are arranged symmetrically such that pairs $(C_0, C_n), (C_1, C_{n-1}), \ldots$ have the same scale and compute waveloss with MSE to ensuring balanced representation across scales.

Instead of directly defining $n$ scales, we specify a shorter ascending sequence $s'_1, \ldots, s'_m$ with $m = \lfloor n/2 \rfloor + 1$, then prepends the reversed one to create symmetrical wave shaped sequence:

$$\{s_1, \ldots, s_n\} = \{s'_m, s'_{m-1}, \ldots, s'_1, s'_2, \ldots, s'_m\}, \quad T_i = T' \times s_i$$

with $T'$ as the base encoder output length.

**Wavescale Loss**   To further enhance the accuracy of the final reconstruction, we introduce a novel loss function *waveloss* denoted as $L_u$. The primary intuition behind waveloss is to enforce consistency across different quantization stages that operate at the same resolution but appear at different points in the hierarchical process. This encourages stage wise coherence when the same temporal resolution is processed at different points, avoiding drift in latent consistency. The waveloss is formally defined as:

$$L_u = \sum_{i=0}^{\lfloor n/2 \rfloor} \left\| \sum_{j=0}^{i} q_j - \sum_{k=0}^{n-i} q_k \right\|_2^2$$

This reduces intra-scale divergence caused by residual noise or interpolation. Statistically, $L_u$ approximates variance minimization among residuals at each resolution:

$$L_u \approx \sum_i \mathrm{Var}(\mathcal{Q}_i) = \sum_i \frac{1}{|\mathcal{Q}_i|} \sum_{q \in \mathcal{Q}_i} \|q - \mu_i\|_2^2, \quad \mu_i = \frac{1}{|\mathcal{Q}_i|} \sum q$$

Minimizing this variance promotes coherent latent distributions and reduces mismatch between down- and upsampling paths. The final loss includes all components:

$$L = \lambda_w L_w + \lambda_f L_f + \lambda_g L_g + \lambda_d L_d + \lambda_{cb} L_{cb} + \lambda_{cm} L_{cm} + \lambda_u L_u$$

This balances reconstruction, perceptual quality, latent consistency, and codebook usage.

# 4 EXPERIMENTS

## 4.1 DATASETS

To ensure generalization across diverse audio domains, we use publicly available datasets. For speech, we include DAPS (Moulines & Charpentier, 1990), DNS Challenge 4 (Dubey et al., 2023), Common Voice (Ardila et al., 2020), and VCTK (Veaux et al., 2017). For music, we use MUSDB (Rafii et al., 2017) and Jamendo (Ramona et al., 2008), and for environmental audio, balanced segments from Audioset (Gemmeke et al., 2017). All audio is resampled to 44 kHz.

We apply stratified sampling as in (Kumar et al., 2023), extracting 1-second training segments uniformly across domains. Validation and test sets use 5- and 10-second clips, respectively. The test set contains 1,000 samples per domain (3,000 total).

## 4.2 TRAINING

Our training setup builds on the DAC framework (Kumar et al., 2023), adopting architectural components proposed in Siuzdak et al. (2024) for improved reconstruction fidelity and stability. Specifically, we follow their design in incorporating 1D convolutions for temporal modeling, depthwise separable convolutions (Howard et al., 2017) to reduce parameter overhead, local attention for efficient contextual modeling, and stochastic noise blocks for latent space regularization and robustness to high dynamic range signals.

Each quantizer contains a codebook with 1024 entries of 64 dimensions. Following the warmup strategy in (Razavi et al., 2019), codebook embeddings that remain unused for the first 1000 training steps are reinitialized with sampled encoder outputs, mitigating early collapse and improving code utilization.

We train for 200k steps using a batch size of 12 on three A6000 GPUs, optimized with AdamW ($\text{lr} = 10^{-4}$, $\beta_1 = 0.8$, $\beta_2 = 0.9$, weight decay $\lambda = 0.999996$). The learning rate is held constant throughout training.

Our loss function is based on the multi-component setup used in Siuzdak et al. (2024), experimentally added two additional losses: waveform L1 loss ($\lambda_w = 0.1$) to stabilize time-domain fidelity and wavescale loss $L_u$ ($\lambda_u = 0.5$) to enforce cross-resolution consistency across quantization levels.

Each full run required approximately 40 GPU-hours. All models were trained under identical conditions to ensure fair comparison.

## 4.3 EVALUATION

Evaluation is performed on 10-second test segments using standard metrics: Mel-spectrogram distance, STFT distance, waveform L1 error, SI-SDR (Le Roux et al., 2019), and FAD (Kilgour et al., 2019). These assess perceptual, spectral, and time-domain fidelity. We also report codebook entropy and effective bitrate based on code usage over time, enabling comparison of compression efficiency across models.

## 4.4 COMPARISON TO OTHER MODELS

**Objective evaluation** As shown in Table 1, among the multiscale models, the *wavescale* variant consistently achieves the best performance across all objective metrics, including Mel spectrogram distance, STFT distance, waveform error, SI-SDR, and FAD. For fair comparison, all WNAC variants (*upscale*, *downscale*, *w/o wavescale loss*) were implemented with identical encoder, decoder, and training settings, differing only in the scaling shape of the quantizer path. Under this controlled setup, the *wavescale* configuration achieves the strongest results, confirming that initiating quantization from high-resolution features and combining it with the proposed waveloss leads to more accurate and perceptually faithful reconstruction. The bitrate efficiency is metric to evaluate effective utilization of the available codebook capacity (Kumar et al., 2023). Among all models, *WNAC (wavescale)* achieves the highest efficiency.

Table 1: Performance Comparison Among Models. *Checkpoint* refers to publicly released weights, while *Trained* indicates models re-trained on our dataset. We compare multiscale RVQ-VAE models including SAT and SNAC. The column *eff.* denotes the bitrate efficiency (%).

| Multiscale RVQ-VAE | | Mel↓ | STFT↓ | WF↓ | SISDR↑ | FAD↓ | eff. (%) ↑ | bitrate (kbps) |
|---|---|---|---|---|---|---|---|---|
| SAT | Checkpoint | 1.438 | 5.636 | 0.037 | 2.287 | 1.604 | - | 4.6 |
| | Trained | 1.415 | 5.419 | 0.043 | -0.478 | 3.318 | 89.15 | |
| SNAC | Checkpoint | 0.797 | 2.020 | 0.035 | 3.725 | 0.754 | - | 2.6 |
| | Trained | 0.853 | 1.859 | 0.038 | 3.132 | 1.302 | 80.33 | |
| WNAC | downscale | 0.872 | 1.891 | 0.033 | 4.879 | 1.248 | 89.26 | 5.2 |
| | upscale | 0.880 | 1.873 | 0.034 | 4.398 | 1.662 | 86.14 | |
| | w/o waveloss | 0.797 | 1.807 | 0.031 | 5.284 | 1.142 | 87.68 | |
| | wavescale | **0.769** | **1.768** | **0.030** | **5.760** | **0.898** | **94.59** | |
| | | 0.831 | 1.806 | 0.034 | 4.383 | 1.080 | 90.68 | 2.52 |

**Subjective evaluation (MUSHRA)** To complement objective metrics, we conducted a MUSHRA listening test with $N = 12$ participants. The comparison included our proposed model and its ablations (SAT, SNAC, downscale, upscale, w/o waveless, WNAC), along with anchors (low-pass 3.5 kHz, 7 kHz) and the hidden reference.

As shown in Fig. 3, the results align with the objective evaluation. The *WNAC* variant achieves the highest perceptual score among models (84.7), approaching the reference (93.3). The *w/o waveloss* and *downscale* variants remain competitive but show degradations, confirming the importance of wavescale quantization. Anchors correctly occupy the lower end of the scale (55–65), validating the reliability of the test design.

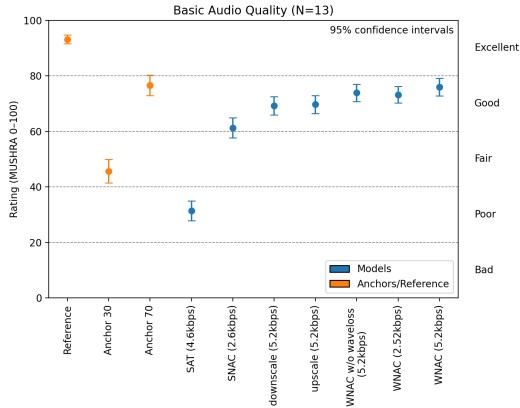

Figure 3: MUSHRA subjective evaluation (mean ± 95% CI, $N = 12$). Models are blue; anchors and reference are orange.

### 4.5 INFERENCE SPEED AND LATENCY

We benchmark the average inference latency of various models using a system equipped with three NVIDIA A6000 GPUs. The results in Table 2 show that WNAC provides two operating points, enabling a flexible trade-off between bitrate, residual depth, and inference efficiency. The low-bitrate WNAC variant (2.52 kbps) uses only 5 residual layers yet achieves SNAC-level latency while maintaining a comparable compression ratio with a smaller codebook (512). In contrast, the 5.2 kbps full-capacity variant employs 15 layers but remains considerably faster than SAT and DAC despite its richer multi-scale representation. These results demonstrate that the downscale–upscale wavescale structure scales efficiently: the deeper

| | Code length | Residual depth | bitrate (kbps) | Time (ms) |
|---|---|---|---|---|
| DAC | ×9 | 9 | 8.0 | 83.6 |
| SAT | ×6.067 | 16 | 4.6 | 71.3 |
| SNAC | ×1.875 | 4 | 2.6 | 21.8 |
| WNAC | ×3.5 | 5 | 2.52 | 24.2 |
| | ×6.04 | 15 | 5.2 | 32.6 |

Table 2: Inference time (ms), code length (relative to encoder output), and residual depth.

Table 3: Ablation Study Results for the Three Domains: Speech, Music, and Environment. For each domain, the performance of the multiscale RVQ-VAE models was compared.

| | | Mel↓ | STFT↓ | WF↓ | SISDR↑ | FAD↓ | bitrate (kbps) |
|---|---|---|---|---|---|---|---|
| Speech | SAT | 1.531 | 6.064 | 0.033 | 0.589 | 4.453 | 4.6 |
| | SNAC | 0.778 | 1.503 | 0.026 | 4.872 | 1.000 | 2.6 |
| | **WNAC** | 0.750 | 1.478 | 0.023 | 6.059 | 0.825 | 2.6 |
| | | **0.698** | **1.447** | **0.020** | **7.611** | **0.523** | 5.2 |
| Music | SAT | 1.472 | 5.553 | 0.028 | 2.059 | 2.964 | 4.6 |
| | SNAC | 0.829 | 1.805 | 0.025 | 4.855 | 1.489 | 2.6 |
| | **WNAC** | 0.803 | 1.700 | 0.023 | 5.840 | 1.138 | 2.6 |
| | | **0.743** | **1.690** | **0.021** | **6.846** | **1.030** | 5.2 |
| Environment | SAT | 1.233 | 4.606 | 0.067 | -4.082 | 4.321 | 4.6 |
| | SNAC | 0.945 | 2.246 | 0.062 | -0.435 | 2.428 | 2.6 |
| | **WNAC** | 0.938 | 2.223 | 0.056 | 1.222 | 2.045 | 2.6 |
| | | **0.855** | **2.143** | **0.049** | **2.784** | **1.756** | 5.2 |

model retains competitive speed, and the lighter model matches SNAC's latency while offering substantially improved representational capacity.

**Bitrate interpretation with code length** Let $L_{\text{enc}}$ denote the encoder output length (latent frames), $c_i$ the relative code length of the $i$-th VQ stage, and $T$ the input duration in seconds. In our formulation, each $c_i$ measures the temporal resolution of the residual entering that quantization stage *relative to* the encoder output resolution (normalized so that full-resolution quantization corresponds to $c_i = 1$). Thus, if a stage downscales its residual before quantization, its code length becomes $c_i < 1$. The total code length reported in Table 2 is

$$c_{\text{total}} = \sum_{i=1}^{n} c_i,$$

which corresponds to the effective number of full-resolution codebooks and directly determines the total number of discrete codes produced by the codec. Therefore, for a fixed encoder–decoder architecture, a smaller $c_{\text{total}}$ implies fewer transmitted indices and thus a higher compression ratio. Given $L_{\text{enc}}$ and a codebook of size $K$, the token rate of the codec and the nominal bitrate becomes

$$r_{\text{tok}} = \frac{L_{\text{enc}}\, c_{\text{total}}}{T} \quad \text{(tokens/s)}, \quad r_{\text{bit}} = r_{\text{tok}} \log_2 K \quad \text{bits/s}.$$

## 4.6 ABLATION STUDY

**Domain Robustness** Audio signals differ in spectral structure across domains: speech is dominated by low-frequency components with fine temporal detail, music spans structured harmonics across the spectrum, and environmental audio is typically broadband and nonperiodic. These variations present distinct reconstruction challenges.

Table 3 shows that the proposed model consistently outperforms SAT and SNAC across domains. It achieves the lowest errors in speech, a clear SI-SDR gain in music, and stable performance in environmental audio where other models degrade.

Figure 4 shows that our model and DAC yield lower and tighter distributions for Mel distance, STFT distance, and waveform error, indicating more consistent reconstruction quality. The com-

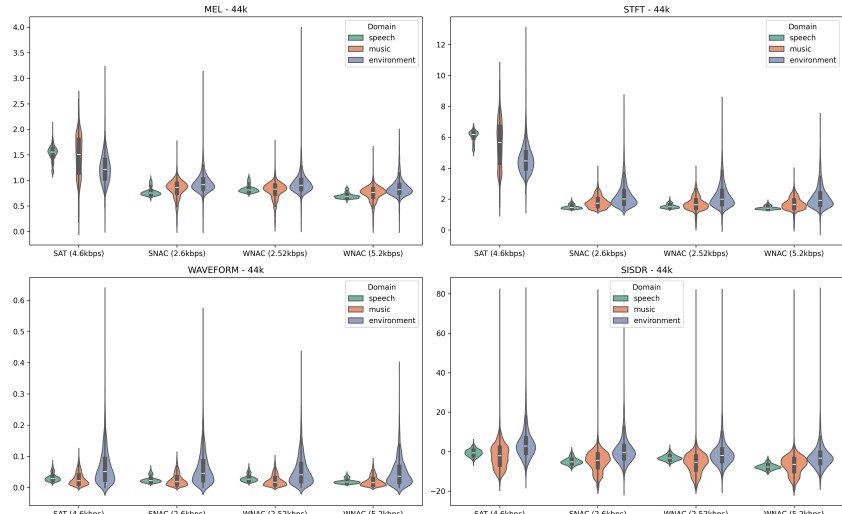

Figure 4: Violin plots of Mel distance, STFT distance, waveform L1 error (lower is better), and SI-SDR (higher is better) across models and domains.

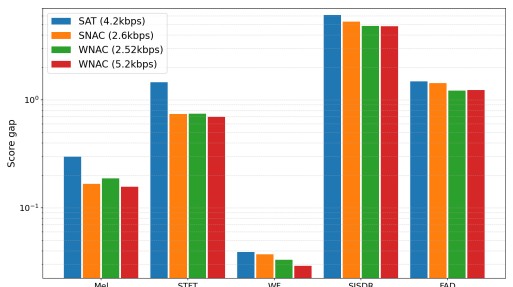

Figure 5: The $(\max - \min)$ gap between *general* and each domain (*speech, music, environment*).

Figure 6: Average unused code ratio across codebook scales (lower is better).

pact violin shapes imply low variance across samples within each domain. In contrast, SAT yields strong median SI-SDR on music but shows broad dispersion and heavy tails, particularly for speech and environmental sounds, implying instability and frequent failures. This inconsistency suggests that SAT may overfit to structured signals like music while lacking robustness to less periodic or noisy inputs. Our model, like DAC, maintains a more stable behavior across all domains, but with improved median performance.

Finally, Figure 5 demonstrates that the proposed model exhibits the smallest domain-wise performance variation, indicating stronger generalization and less domain-specific overfitting.

**Evaluation at Low Bitrate Settings**    To fairly compare WNAC against other models in an high compressing regime, we configured a lightweight WNAC variant operating at 2.25 kbps which is slightly *lower* than SNAC's 2.6 kbps. SNAC uses four RVQ stages with average-pooling-based upscaling, which enforces residual temporal resolutions that scale strictly in powers of two. This architectural constraint makes deep multiscale RVQ difficult to stabilize at low bitrates.

In contrast, WNAC allows arbitrary scale assignments through the wavescale (fine → coarse → fine) structure. However, when naively compressing WNAC to SNAC's bitrate, the fixed endpoints at scale = 1 force the intermediate scales to shrink too aggressively, degrading performance. To avoid this, we increased the number of quantizers while reducing the codebook size from 1024 to

512 (2.52 kbps) and 256 (2.25 kbps), then assigned scale factors that mimic the effective resolutions induced by SNAC's adaptive pooling pipeline (e.g., $1 \rightarrow 0.5 \rightarrow 0.25 \rightarrow 0.5 \rightarrow 1$), thereby ensuring a fair comparison and preserving the intended behavior of the wavescale hierarchy.

Table 4: Comparison of WNAC and SNAC under matched low-bitrate conditions. WNAC operates at a lower bitrate (2.25 kbps, 2.52 kbps) yet achieves consistently better reconstruction quality and significantly higher codebook efficiency.

| Method | Mel↓ | STFT↓ | WF↓ | SISDR↑ | FAD↓ | eff.(%)↑ | bitrate (kbps) |
|--------|------|-------|-----|--------|------|----------|----------------|
| SNAC | 0.853 | 1.859 | 0.038 | 3.132 | 1.302 | 80.33 | 2.60 |
| WNAC | 0.845 | 1.829 | 0.035 | 3.988 | 1.136 | **91.70** | 2.25 |
| | **0.831** | **1.806** | **0.034** | **4.383** | **1.080** | 90.68 | 2.52 |

Table 4 shows that WNAC achieves *better reconstruction metrics on all measures* (Mel, STFT, WF, SISDR, FAD) while operating at a *lower* bitrate. Notably, WNAC attains about **90%** codebook efficiency, a substantial improvement over SNAC's **80.33%**, indicating healthier utilization and more uniform distribution of residual information across stages.

These results demonstrate that WNAC retains its structural advantages even in the highly bandwidth-constrained 2–3 kbps regime, outperforming SNAC both in reconstruction quality and codebook efficiency.

**Codebook Utilization**   To evaluate the effectiveness of our architecture, we analyzed codebook utilization across quantization scales. While lower scales typically suffer from inefficient code usage due to coarse residuals, Figure 6 shows that our model achieves significantly higher utilization at these stages compared to baseline variants.

This indicates that the Wavescale structure and waveloss together promote richer and more consistent codebook usage, improving compression efficiency and latent expressivity where conventional approaches fall short (Zeghidour et al., 2021; Borsos et al., 2022).

Further ablations are in Appendix D, covering bitrate efficiency across domain (Table 5), scale-wise alignment (Fig. 9), waveloss weight evaluation, (Table 6), early–stage reconstructions (Tables 7, 8, Fig. 10), downstream WER (Fig. 11), and latent visualizations (Fig. 12).

## 5   CONCLUSION

We presented the Wavescale Neural Audio Codec (WNAC), a multiscale residual vector quantization framework that replaces the conventional bottom-up (coarse→fine) hierarchy with a down-scale–upscale design. Together with a scale-aware loss, this structure enables more accurate preservation of both low- and high-frequency content and reduces information loss in early quantization stages. Across multiple datasets and evaluation settings, WNAC consistently outperforms state-of-the-art multiscale RVQ models in both objective and perceptual metrics, demonstrating its robustness and generalization capability.

**Limitations and Future Work**   Although WNAC provides higher-quality and more predictable discrete representations, its fine→coarse→fine ordering differs from the coarse-first assumption underlying existing next-scale prediction frameworks such as SAT and SNAC. As a result, these hierarchical predictors cannot be used directly without modifying their conditioning structure. While token-level autoregressive modeling remains fully compatible, a "wavescale-aware" hierarchical predictor that aligns with WNAC's quantization flow is a promising direction for future work.

Our focus in this paper is on improving the quality and efficiency of the quantized representations themselves. Nevertheless, Appendix D.5.2 includes a lightweight autoregressive diagnostic showing that WNAC produces significantly more predictable codes than SNAC, suggesting that dedicated wavescale-aware generative models are both feasible and potentially highly effective.

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

# A  RVQ PROPERTIES OF MODELING LOW-FREQUENCY CONTENT IN EARLY STAGES

Residual vector quantization (RVQ) exhibits a well-known frequency ordering effect: early quantizers predominantly model low-frequency (LF) components, while later quantizers focus increasingly on high-frequency (HF) residuals. Below, we summarize the mechanism behind this phenomenon and explain why it leads to LF loss in coarse-first multiscale RVQ systems.

**High-pass nature of residual computation.** In a conventional coarse-to-fine RVQ, the first quantizer learns an approximation $q_1$ of the full-resolution input $x$. Since quantization error is minimized in an $\ell_2$-optimal manner, the learned codebook tends to represent the largest-variance components of the signal: typically the slowly varying, low-frequency structure.

As a result, the residual after the first quantization step,

$$r_1 \; = \; x - q_1,$$

behaves similarly to a high-pass filtered version of $x$: most LF energy is already removed by $q_1$, leaving residual energy concentrated in higher frequencies. Any LF mismatch in $q_1$ becomes unrecoverable, because all subsequent stages operate only on $r_1$, which contains no meaningful LF components.

**Consequences for multiscale RVQ.** Since later RVQ stages receive increasingly high-pass residuals, they specialize in modeling HF detail. This progression is beneficial for speech compression but becomes problematic for music and environmental audio, whose LF regions contain harmonics, chords, ambience, and other semantically rich content.

This explains why coarse-first multiscale RVQ codecs frequently exhibit:

- LF attenuation or collapse in early stages,

- poor LF consistency across scales,

- cascading quantization error when LF structure is mismatched.

**WNAC's fine-first solution.** WNAC inverts the coarse-first hierarchy by quantizing fine-resolution features before any residual subtraction is performed. By doing so, both low- and high-frequency components are explicitly encoded in the early quantizers, rather than being removed by a coarse initial approximation.

The subsequent coarse-to-fine upscaling stages refine global structure rather than attempting to reconstruct missing LF content. This reordering reduces cumulative quantization error, because early quantizers operate directly on the high-variance (full-spectrum) components that dominate reconstruction loss.

# B  DETAILS OF DATASETS

In this section, we describe the domain-specific characteristics of the datasets used in our experiments. Understanding the spectral and periodic properties of each domain provides important context for interpreting the reconstruction performance across speech, music, and environmental sounds.

## B.1  SPEECH DOMAIN

We use the DAPS, DNS Challenge 4, Common Voice, and VCTK datasets to represent speech audio. Speech signals are dominated by low- to mid-frequency formant structures and exhibit moderate periodicity primarily within the low quefrency range(0-5ms). Due to the relatively simple spectral organization and limited fine-grained harmonic content, speech is easier to compress and reconstruct with multiscale RVQ architectures, leading to consistently high performance across Mel distance, STFT distance, and waveform error metrics.

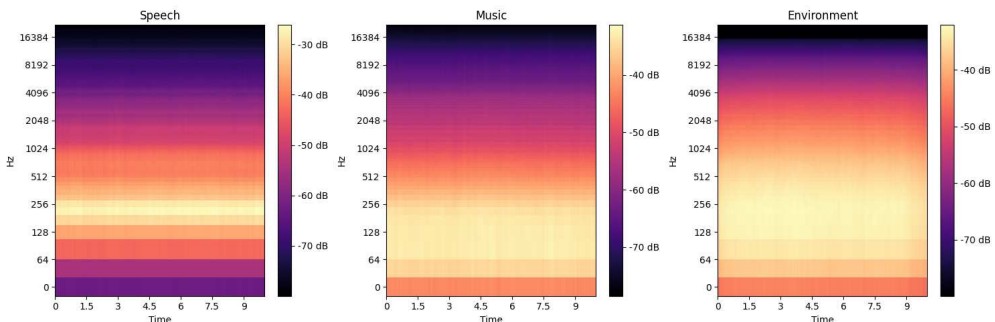

Figure 7: Average log-magnitude spectrograms for speech, music, and environmental domains. Speech exhibits concentrated energy in formant-related low-mid frequencies; music distributes energy more broadly with harmonic richness; environmental sounds show flat, noise-like spectral characteristics.

## B.2 MUSIC DOMAIN

The music domain includes samples from the MUSDB and Jamendo datasets, covering a wide variety of genres, instruments, and mixing styles. Music signals are structurally more complex, containing dense harmonic structures and rich transient patterns extending across low, mid, and high frequency ranges. Although music exhibits stronger periodicity compared to speech, the fine-grained nature of its harmonic overtones increases the difficulty of compression and reconstruction, often resulting in slightly lower objective metric scores.

## B.3 ENVIRONMENTAL DOMAIN

Environmental audio is drawn from the balanced train set of Audioset. Unlike speech or music, environmental sounds generally lack stable harmonic structures, especially in high-frequency regions. Their noise-like, unstructured composition makes it challenging for multiscale RVQ to model residuals effectively at deeper scales, leading to larger reconstruction errors and lower performance in perceptual and spectral metrics.

## B.4 DOMAIN-SPECIFIC ANALYSIS

We further analyze the spectral and periodic characteristics of speech, music, and environmental audio from test dataset to explain their domain-specific differences in reconstruction performance.

As shown in Figure 7, the average log-magnitude spectrograms reveal distinct energy distribution patterns across domains. Speech signals exhibit strong energy concentrations in the low-to-mid frequency range (approximately 200–3000 Hz), corresponding to stable formant structures produced by vocal tract resonances. Music signals show a broader and denser distribution of energy across the spectrum, driven by harmonic complexity and overlapping instruments. Environmental sounds, in contrast, exhibit relatively flat and noise-like spectral profiles, indicating a lack of dominant tonal structure.

To further characterize temporal regularity, Figure 8 presents violin plots of quefrency peak distributions extracted from five frequency bands. Speech shows tight, low-quefrency peaks (typically 3–6 ms) in low and mid bands, corresponding to pitch and formant-related periodicity. This strong, compact periodic structure supports efficient residual quantization and leads to lower reconstruction errors. Music demonstrates broader and more skewed periodic distributions, especially in mid-to-high bands, due to richer harmonic structures and transient elements. In contrast, environmental audio exhibits flat, dispersed quefrency distributions across all bands, indicating minimal periodicity and high variability. This lack of temporal structure makes residual vector quantization less effective, resulting in consistently higher reconstruction errors across all objective metrics.

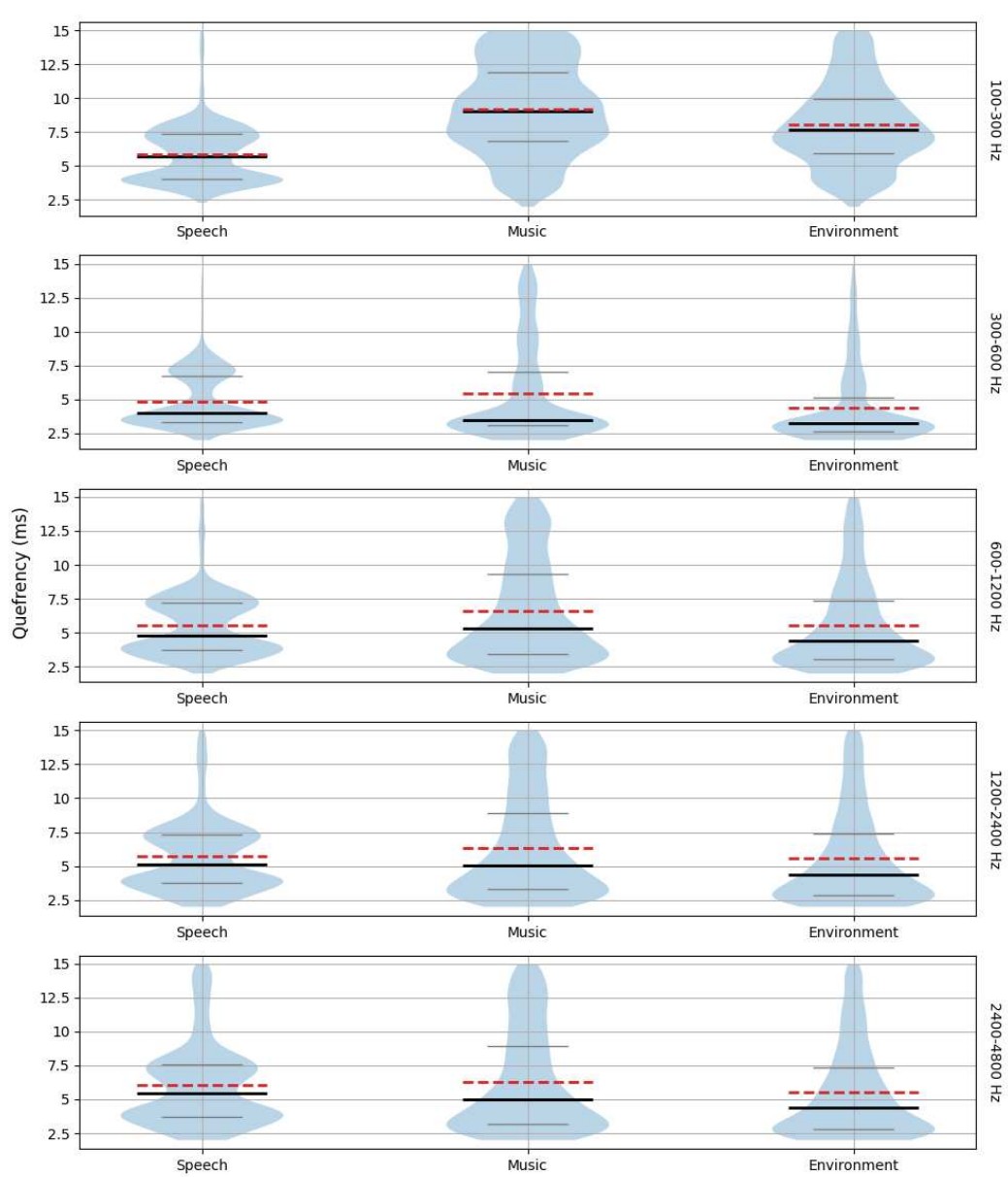

Figure 8: Violin plots of quefrency peak distributions across five frequency bands for each domain. Red dashed lines indicate the mean, thick black lines indicate the median, and thin gray lines represent the first and third quartiles (Q1, Q3). These plots summarize domain-dependent periodicity patterns that influence multiscale residual quantization performance.

Table 5: Bitrate efficiency comparison across domains. Higher values indicate more efficient use of codebook tokens under a fixed bitrate.

|  | General | Speech | Music | Environment |
|---|---|---|---|---|
| upscale | 86.14 | 85.316 | 83.070 | 84.845 |
| w/o waveloss | 90.677 | 91.340 | 89.616 | 90.084 |
| wavescale | **94.59** | **91.82** | **90.64** | **92.57** |

Together, the spectral and quefrency-based analyses provide complementary evidence that structured, periodic audio (as in speech) is easier to compress and reconstruct than complex (music) or unstructured (environmental) signals.

## C    DETAILS OF IMPLEMENTATIONS

**Hardware Setup.**    All experiments were conducted on a single node equipped with an AMD EPYC 7763 64-Core Processor (128 threads) and 503 GB RAM. The node was configured with three NVIDIA A6000 GPUs (48 GB VRAM each) connected via PCIe. Storage consisted of a 3.5 TB NVMe SSD mounted at `/data1` to enable high-speed data access.

**Software Environment.**    The system ran Ubuntu 22.04.4 LTS with Linux kernel 5.15. CUDA 12.4 and cuDNN 9.1.0 were used for GPU acceleration. Python 3.12.8 and PyTorch 2.5.1+cu124 were employed for model development and training, along with supporting libraries including NumPy 1.26, SciPy 1.12, Matplotlib 3.8, and librosa 0.10.1.

**Training Settings.**    Models were trained using the AdamW optimizer with an initial learning rate of $1 \times 10^{-4}$, $\beta_1 = 0.8$, $\beta_2 = 0.9$, and a weight decay of $1 \times 10^{-4}$. A linear learning rate decay with a multiplicative factor of 0.999996 per step was applied. Training proceeded for 400,000 iterations with a batch size of 12, distributed across three GPUs using DistributedDataParallel (DDP). Mixed-precision training (Automatic Mixed Precision, AMP) was enabled to optimize memory usage and computational throughput.

**Inference Settings.**    Inference was performed with a batch size of 1 on a single A6000 GPU, evaluating audio signals at both 22 kHz and 44 kHz sample rates. Latency measurements were collected under identical hardware conditions without additional quantization or model pruning.

**Reproducibility.**    All    experiments    were    conducted    with    a    fixed    random    seed of    0.    CUDA    deterministic    modes    were    enabled    where    applicable    by    setting `torch.backends.cudnn.deterministic = True` and disabling benchmarking via `torch.backends.cudnn.benchmark = False`.

## D    ADDITIONAL EXPERIMENTS

To complement the main results, we include a set of additional experiments that further analyze the behavior, efficiency, and design decisions of the proposed Wavescale Neural Audio Codec. These experiments provide deeper insights into the internal mechanisms and performance trade-offs of our approach.

### D.1    CODEBOOK UTILIZATION ANALYSIS

To assess the efficiency of token usage across different audio types, we compute bitrate efficiency scores following the methodology proposed in DAC. This score reflects how effectively the quantized codebooks capture information under a fixed bitrate budget, with higher values indicating more compact and expressive representations.

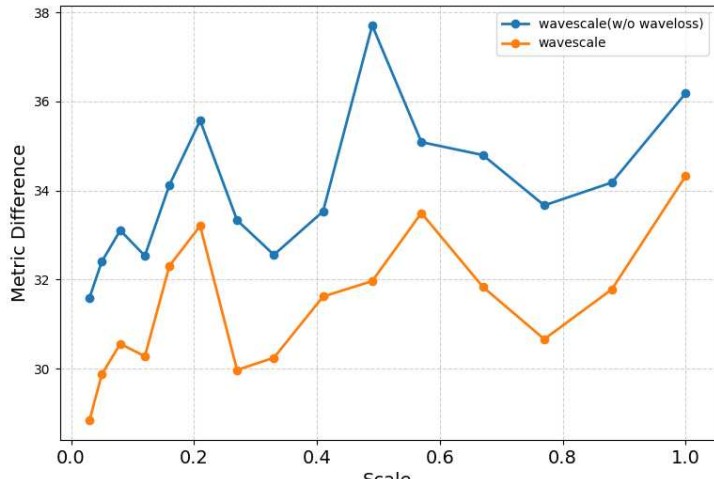

Figure 9: Scale-wise reconstruction difference comparison between models trained with and without waveloss. Lower metric differences indicate better reconstruction quality. Waveloss provides consistent improvements across scales, particularly at intermediate resolutions.

Table 5 reports the bitrate efficiency across four settings—General (mixed-domain), Speech, Music, and Environment. Compared to both the scale-based baseline and the variant without waveloss, the proposed Wavescale model consistently achieves the highest efficiency across all domains.

This demonstrates that our architecture not only improves reconstruction quality but also encodes information more economically, reducing redundancy in the quantized representation. In particular, we observe significant gains in the music and environmental domains, which contain highly structured or diverse spectral features. This suggests that the proposed model better adapts to domain-specific characteristics, leading to more efficient usage of available codebook capacity.

## D.2 WAVELOSS ABLATION BY STAGE

To further investigate how waveloss contributes across different quantization stages, we analyze the scale-wise reconstruction quality differences between models trained with and without waveloss. For each scale, we partially reconstruct the signal by accumulating quantized vectors up to that stage and compute the reconstruction metrics.

Figure 9 shows the average metric difference at each scale. The model trained with waveloss consistently achieves lower reconstruction error across almost all scales. In particular, the improvement is more pronounced at mid-level scales, indicating that cross-scale consistency enforced by waveloss is especially beneficial when refining intermediate-resolution representations.

These findings highlight that waveloss not only improves the final reconstruction quality, but also systematically enhances the stability and expressiveness of intermediate quantization stages, leading to more accurate and robust multi-stage residual reconstruction.

## D.3 WAVELOSS ABLATION BY HYPERPARAMETER

Table 6 shows that a moderate waveloss weighting ($\lambda = 0.5$) achieves the best overall performance across distortion (Mel, STFT, WF) and perceptual (FAD) metrics. Larger values (e.g., $\lambda = 10.0$) lead to degraded reconstruction quality, while smaller values (e.g., $\lambda = 0.1$) result in weaker perceptual consistency. These results highlight the importance of selecting an appropriate weighting coefficient within a low range (e.g., $\lambda \in [0.1, 1.0]$) to balance scale-wise consistency and reconstruction fidelity.

Table 6: Effect of waveloss weighting coefficient $\lambda$ on reconstruction performance. Optimal results are observed at $\lambda = 0.5$.

| $\lambda$ | Mel↓ | STFT↓ | WF↓ | SISDR↑ | FAD↓ |
|---|---|---|---|---|---|
| 0.1 | 0.780 | 1.771 | **0.030** | 5.709 | 0.996 |
| 0.5 | **0.769** | **1.768** | 0.030 | **5.760** | **0.898** |
| 1.0 | 0.772 | 1.777 | **0.030** | 5.599 | 0.911 |
| 2.0 | 0.773 | 1.776 | 0.031 | 5.738 | 0.996 |
| 10.0 | 0.792 | 1.789 | 0.031 | 5.572 | 0.963 |

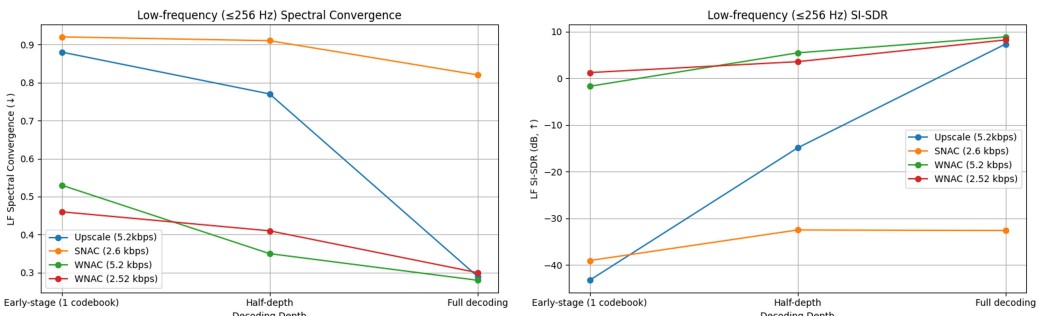

Figure 10: Low-frequency ($\leq$256 Hz) reconstruction metrics across decoding depths.

## D.4 ABLATION ON EARLY-STAGE QUANTIZATION IN WNAC

To investigate whether the early quantization stages in the proposed WNAC retain richer semantic information, we conducted an ablation experiment under two settings: Table 7 reports reconstruction quality using only the first residual VQ stage, and Table 8 shows results when decoding with the first half of the residual VQ stages.

To further analyze this phenomenon in the low-frequency band, Figure 10 plots low-frequency ($\leq$256 Hz) metrics for Upscale RVQ, SNAC, and two versions of WNAC (5.2 kbps and 2.52 kbps) under three decoding depths: early-stage (1 codebook), half-depth, and full decoding.

In the early-stage setting, the Upscale baseline shows severe low-frequency collapse, with a strongly negative band-limited SI-SDR (about $-43.25$ dB) and high LF spectral convergence (about $0.88$), indicating that its first residual subtraction removes most of the slowly varying components. SNAC performs better but still retains substantial low-frequency error. In contrast, both WNAC variants preserve significantly more LF structure: the 5.2 kbps model achieves high band-limited SI-SDR (about $-1.7$ dB) and low LF spectral convergence (about $0.53$), while the 2.52 kbps model shows similar behavior with slightly weaker but still substantially better LF reconstruction than Upscale and SNAC.

A similar pattern appears in the half-depth setting. Upscale RVQ continues to struggle with restoring low-frequency structure, whereas both WNAC models maintain LF performance close to their full-depth results. Even at the lower bitrate, WNAC (2.52 kbps) retains stable low-frequency behavior, outperforming Upscale RVQ by a large margin.

Finally, in the full-decoding setting, both WNAC variants achieve strong LF spectral convergence and SI-SDR, consistently outperforming Upscale RVQ and SNAC. The presence of both bitrates demonstrates that WNAC's fine-to-coarse quantization pathway consistently preserves low-frequency structure across different bitrate operating points, and that this behavior is not dependent on model capacity alone. Importantly, this also indicates that accurate preservation of low-frequency content in the earliest stages has a lasting influence on the final reconstruction quality achieved at full decoding.

Table 7: Reconstruction quality when only the first residual VQ stage is used.

| Model | scale conf. | Mel ↓ | STFT ↓ | WF ↓ | SISDR ↑ | FAD ↓ | bitrate (kbps) |
|-------|-------------|-------|--------|------|---------|-------|----------------|
| SAT | upscale | 2.158 | 6.185 | 0.059 | -9.774 | 11.041 | 4.6 |
| SNAC | upscale | 2.195 | 3.256 | 0.087 | -42.807 | 7.850 | 2.6 |
| WNAC | upscale | 3.640 | 5.111 | 0.090 | -49.571 | 20.817 | 5.2 |
| | w/o waveloss | 1.706 | 2.836 | 0.068 | -7.27 | 6.130 | 5.2 |
| | wavescale | **1.138** | **2.172** | **0.050** | **-1.083** | **2.319** | 2.52 |
| | | 1.322 | 2.391 | 0.057 | -3.454 | 3.840 | 5.2 |

Table 8: Reconstruction quality when using the first half of the residual VQ stages.

| Model | scale conf. | Mel ↓ | STFT ↓ | WF ↓ | SISDR ↑ | FAD ↓ | bitrate (kbps) |
|-------|-------------|-------|--------|------|---------|-------|----------------|
| SAT | upscale | 1.534 | 5.724 | 0.046 | -1.410 | 2.847 | 4.6 |
| SNAC | upscale | 1.691 | 2.799 | 0.082 | -31.244 | 6.190 | 2.6 |
| WNAC | upscale | 1.270 | 2.298 | 0.083 | -25.268 | 5.439 | 5.2 |
| | w/o waveloss | 0.855 | 1.843 | 0.038 | 2.914 | 1.346 | 5.2 |
| | wavescale | 0.975 | 1.971 | 0.044 | 0.891 | 1.738 | 2.52 |
| | | **0.850** | **1.848** | **0.038** | **2.798** | **1.332** | 5.2 |

## D.5 DOWNSTREAM TASK

### D.5.1 WER EVALUATION ON COMMON VOICE

We further evaluated the proposed WNAC in a downstream automatic speech recognition (ASR) task. WER (Word Error Rate, lower is better) was computed using 300 utterances randomly sampled from the `Common Voice` dataset. For ASR inference, we used the `speech_recognition` library with the `recognize_google` backend[2], following prior work that evaluates codecs by feeding reconstructed audio directly into an off-the-shelf recognizer.

To isolate the impact of the proposed *waveloss*, we performed an ablation within WNAC and compared it to other models. The evaluation considered reconstructions using only the first residual quantizer, half of the residual quantization depth, and all residual layers.

Figure 11 shows that using only a single residual quantizer drives all systems to a WER of 1.00, indicating that such extreme compression removes sufficient linguistic content for ASR. At half residual depth, WNAC attains 0.58 WER (tied with its ablation without waveloss) and clearly outperforms SAT (0.70), Upscale (0.88), and SNAC (0.98), suggesting that the proposed wavescale quantization better preserves phonetic cues under partial reconstruction. With all residual layers enabled, WNAC achieves the lowest WER of 0.51, improving over SAT (0.56), Upscale (0.53), and SNAC (0.61), and slightly surpassing its waveloss ablation (0.52). Overall, the dominant downstream ASR gains stem from the wavescale quantization path, while waveloss contributes a small but consistent additional improvement at full depth.

---

[2]Specifically, we used `recognizer.recognize_google(audio, language="en-US")`, which invokes Google's production ASR service.

### D.5.2 AUTOREGRESSIVE PREDICTABILITY ANALYSIS OF WNAC CODES

To evaluate whether WNAC provides more autoregressively predictable discrete representations than SNAC, we analyze a lightweight Audio AutoRegressive (AAR) model trained on pretrained SNAC (2.6 kbps) and WNAC (2.52 kbps) codecs. All experiments use 2-second audio excerpts and identical CLAP conditioning.

At each codebook stage $i$, the codec outputs a discrete code $c_i$, which is mapped to a latent embedding $\ell_i$ via that stage's codebook. Because residual vector quantization represents the signal using a cumulative sum of stage-wise residuals, the effective latent after stage $i$ is

$$\tilde{\ell}_i = \sum_{j=0}^{i} \ell_j.$$

The AAR receives this cumulative latent $\tilde{\ell}_i$ at each step and predicts the next-stage code $c_{i+1}$ under teacher forcing while training. For SNAC (coarse-to-fine), stage order is natural. For WNAC (fine→coarse→fine), we order the stages by increasing temporal resolution (smallest scale first), allowing cumulative decoding to progress from low- to high-resolution latents.

To compare predictability of the trained models independently of differing codebook sizes (512 for WNAC vs. 4096 for SNAC), we use the following codebook-invariant metrics:

- **NCE (Normalized Cross-Entropy)**: Conditional CE normalized by the empirical marginal entropy of the codes.
- **NACC (Normalized Accuracy)**: Accuracy normalized against random chance ($1/K$), reflecting how much explainable structure the AAR captures.
- **MI(b)**: Estimated mutual information in bits/token between AAR inputs and target codes.

| Model | NCE ↑ | NACC ↑ | MI (bits) ↑ | ΔMI (normal–rand) ↑ |
|---|---|---|---|---|
| WNAC (2.52 kbps) | **0.307** | **0.207** | **2.642** | **3.414** |
| SNAC (2.6 kbps) | 0.222 | 0.113 | 2.495 | 2.564 |

Table 9: AAR predictability comparison of WNAC vs. SNAC using codebook-size-invariant metrics. ΔMI indicates the drop in mutual information when AAR inputs are randomized (measuring how much usable temporal structure is present in the codec's latent space).

**Discussion.** Even when evaluated using a compact AAR (due to time constraints), WNAC exhibits substantially higher predictability across all codebook-independent metrics: NCE (+65% relative), NACC (+73% relative), and MI (+0.34 bits/token). Moreover, the randomization gap ΔMI is more than twice as large for WNAC, indicating that WNAC codes encode significantly more condition-dependent structure that an autoregressive model can exploit.

Although full generation quality evaluation is beyond the scope of this diagnostic experiment, these results show that WNAC provides richer and more predictable discrete representations for AR modeling compared to SNAC, despite operating at a similar bitrate.

### D.6 COMPARISON OF SINGLE-SCALE RVQ MODEL AND MULTISCALE RVQ MODELS

Table 10 compares the proposed WNAC against both single-scale (DAC) and multiscale RVQ models (SAT, SNAC). While DAC achieves strong reconstruction quality, it requires a substantially higher bitrate (8.0 kbps). In contrast, multiscale RVQ models operate at significantly lower bitrates but often suffer from degraded low-frequency fidelity or unstable codebook utilization.

Among the multiscale baselines, SAT exhibits the lowest bitrate but shows large distortions in all perceptual metrics and a negative SI-SDR, indicating severe reconstruction instability. SNAC improves reconstruction quality but still falls short in FAD and efficiency.

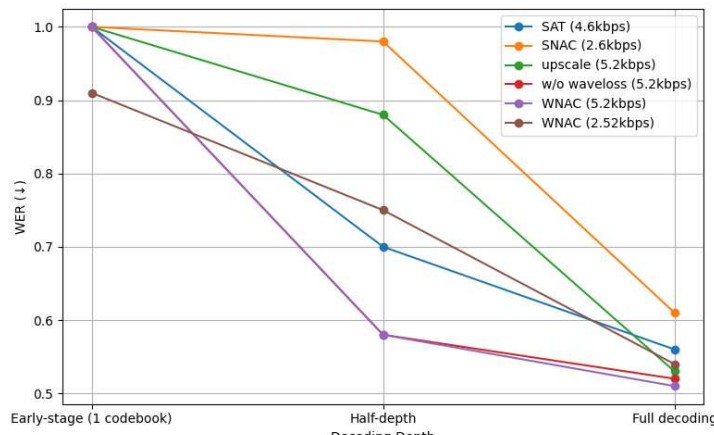

Figure 11: Downstream WER (Word Error Rate) on Common Voice utterances. Evaluation compares reconstructions using different residual depths. WNAC with *waveloss* achieves the best performance.

| Model | scaling shape | Mel ↓ | STFT ↓ | WF ↓ | SI-SDR ↑ | FAD ↓ | Eff. (%) ↑ | bitrates (kbps) |
|---|---|---|---|---|---|---|---|---|
| DAC | single-scale | 0.570 | 1.536 | 0.020 | 10.657 | 0.356 | 87.468 | 8.0 |
| SAT | | 1.415 | 5.419 | 0.043 | -0.478 | 3.318 | 89.15 | 4.6 |
| SNAC | upscale | 0.853 | 1.859 | 0.038 | 3.132 | 1.302 | 80.33 | 2.6 |
| WNAC | | 0.880 | 1.873 | 0.034 | 4.398 | 1.662 | 86.14 | 5.2 |
| WNAC | wavescale | 0.769 | 1.768 | 0.030 | 5.760 | 0.898 | 94.59 | 5.2 |
| | | 0.831 | 1.806 | 0.034 | 4.383 | 1.080 | 90.68 | 2.52 |

Table 10: Comparison of single-scale and multiscale RVQ models in reconstruction metrics, codebook efficiency, and bitrate. Lower Mel/STFT/WF/FAD indicate better reconstruction fidelity; higher SI-SDR and efficiency indicate better performance.

WNAC outperforms both SAT and SNAC across all metrics while maintaining a moderate bitrate of 5.2 kbps. Furthermore, the proposed wavescale variant yields the best overall performance: it achieves the lowest Mel/STFT/WF errors, the highest SI-SDR, and the lowest FAD, while also attaining the highest codebook efficiency (94.6%). This demonstrates that WNAC reduces the typical drawbacks of multiscale RVQ while retaining the compression benefits of a multiscale design.

Overall, WNAC provides a favorable trade-off between bitrate and reconstruction quality, outperforming existing multiscale approaches and achieving competitive fidelity relative to a single-scale codec at substantially lower bitrate.

### D.7 LATENT VISUALIZATION

To further validate the consistency of information modeling across residual depths, we visualize spectrogram difference maps for the music domain under three multiscale RVQ architectures: downscale-only, upscale-only, and the proposed Wavescale structure. The spectrogram difference maps are computed by measuring the magnitude differences between adjacent scale groups.

As shown in Figure 12, the Wavescale model maintains consistently moderate and stable spectrogram difference patterns across all quantization stages (e.g., Scale 0–2, 2–4, ..., 10–12). In contrast, the downscale-only and upscale-only models exhibit some fluctuations in earlier stages and show uneven refinement behavior across different depths.

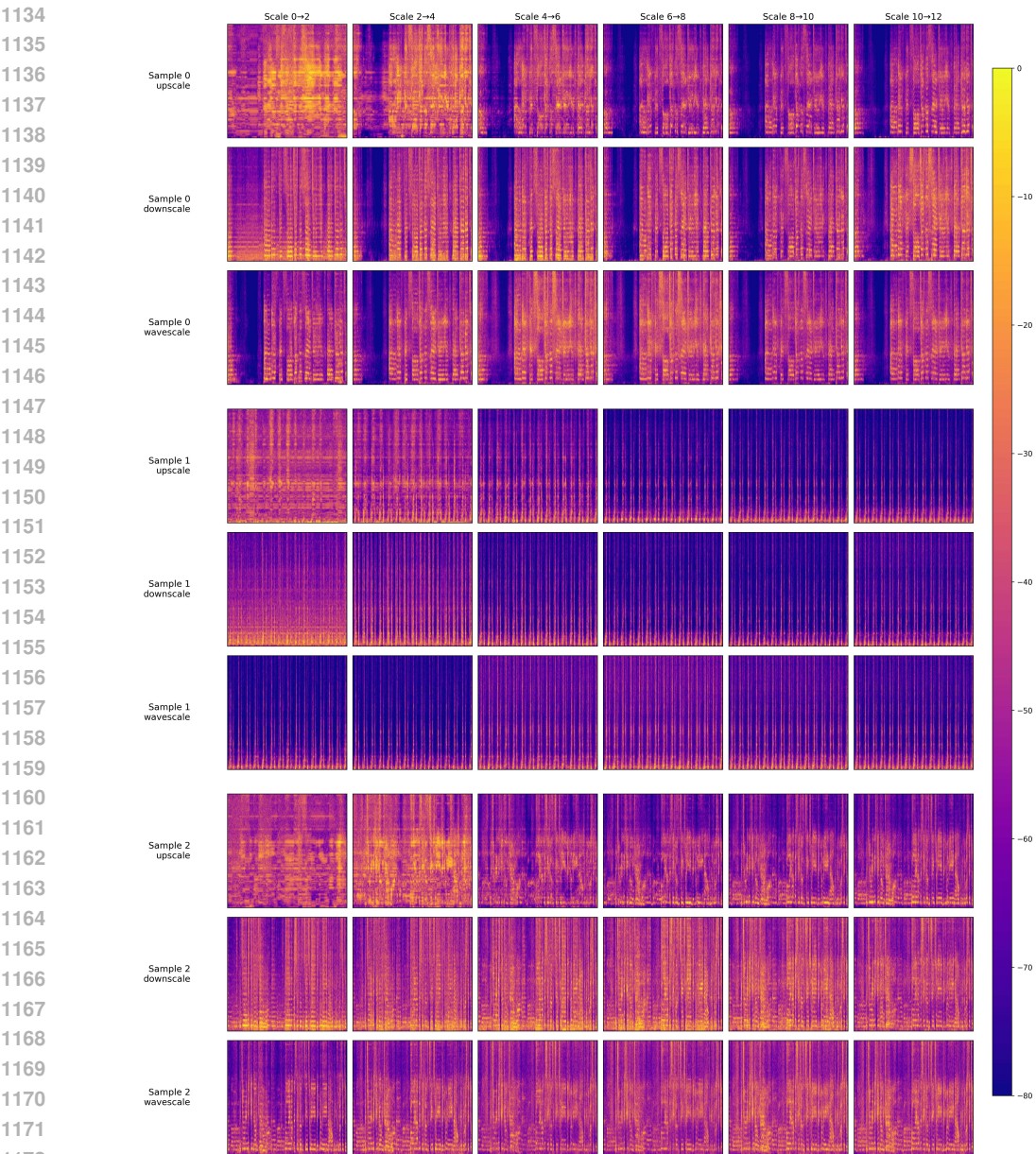

Figure 12: Scale-wise spectrogram difference maps across three models (upscale-only, downscale-only, and the proposed wavescale) for multiple audio samples.

These observations indicate that the Wavescale architecture promotes smoother and more gradual integration of information throughout the multiscale quantization hierarchy. This property is crucial for effectively reconstructing complex audio domains such as music, where information is distributed across multiple temporal and spectral scales.

Overall, these results reinforce that the Wavescale structure enables more consistent residual modeling across all depths, leading to enhanced reconstruction stability compared to conventional downscale- or upscale-only approaches.

# E BROADER IMPACTS

This work introduces a neural audio codec trained on publicly available datasets spanning speech, music, and environmental sounds. No private or personally identifiable information is used, and all datasets are commonly adopted in academic research. The model is designed for compression and reconstruction, and is not explicitly trained for generative or surveillance tasks. However, as with many discrete representation models, the resulting tokens could potentially be integrated into generative pipelines, raising considerations around voice synthesis or unauthorized audio replication. While our work does not explore or enable such use cases, we acknowledge their possibility in downstream applications.

On the positive side, efficient audio coding can benefit communication systems in bandwidth-constrained settings, enabling broader access to high-fidelity audio. We evaluate domain robustness to minimize performance bias across audio types and encourage responsible use of the model. We believe this work presents minimal ethical or societal risks in its current form, but we support continued discussion around safeguards and transparency in neural audio technologies.

