# OpenReview forum: "Wavescale Neural Audio Codec: Bidirectional Multiscale Residual Quantization for High-Fidelity Audio Compression"
_ICLR.cc/2026/Conference — ICLR 2026 Conference Desk Rejected Submission_

### Official Review · Reviewer_Cbcn · 2025-10-29

**Soundness:** 3
**Presentation:** 3
**Contribution:** 3
**Rating:** 6
**Confidence:** 2

**Summary:**

The paper introduces a novel neural audio codec, termed the Wavescale Neural Audio Codec (WNAC), which aims to address the limitations of existing high-fidelity audio compression methods. The central motivation stems from a key limitation of contemporary multiscale Residual Vector Quantization (RVQ) models (e.g., SAT, SNAC), which typically employ a unidirectional coarse-to-fine upscaling hierarchy. The authors persuasively argue that this assumption—that coarse, low-frequency information can be adequately captured in early, low-resolution stages—often fails for non-speech audio such as music and environmental sounds, where fine details can be present in low-frequency components, leading to information loss.

**Strengths:**

Clear and Well-Supported Motivation: The paper provides a compelling critique of the standard coarse-to-fine paradigm in multiscale audio codecs, supported by domain-specific analysis showing its failure on non-speech audio.

Novel and Well-Tailored Methodology: The proposed Wavescale architecture, a symmetric "downscale-then-upscale" flow, is a highly innovative solution. It is effectively complemented by a scale-aware waveloss that enforces representational consistency and improves reconstruction stability.

Rigorous and Convincing Experiments: The experimental design systematically evaluates the method across diverse domains. The results convincingly demonstrate SOTA performance on main benchmarks (Table 1), strong subjective listening scores (Figure 3), and the necessity of each core component through ablation studies.

High-Quality Presentation and Downstream Validation: The paper is well-written with intuitive visualizations (e.g., Figure 1). The validation on a downstream ASR task (Table 8) further strengthens the claim of producing useful representations.

**Weaknesses:**

Unclear Low-Frequency Preservation Mechanism: The paper claims the initial fine-scale processing preserves low-frequency information (Line 29), which is counter-intuitive and lacks a clear technical explanation.

Incompatibility with Autoregressive Generation: The proposed top-down quantization flow breaks the assumptions of existing next-scale prediction methods (e.g., in SAT, SNAC), limiting the codec's immediate applicability for popular autoregressive generation tasks.

Notational Ambiguity in Loss Function: The summation symbols in the VQ-VAE loss equations for L_cb and L_cm (Lines 192-193) lack indices, which is a minor but important notational error.

**Questions:**

Following up on the point raised in Weakness #1, could the authors elaborate on the mechanism by which the initial fine-resolution stages of WNAC effectively preserve low-frequency information, which typically dominates the signal energy? Is this an inherent property of the encoder architecture, or is there a specific design element that facilitates this?

Regarding the waveloss, which is computed between q_i and q_{n-i}, does this imply that the total number of quantization stages n must be odd to have a central pivot? How are cases with an even n handled? Furthermore, did the authors experiment with alternative consistency-enforcing losses, such as a cycle-consistency loss or a direct constraint on the latent residuals z?

In the downstream ASR evaluation (Table 8), was the ASR model applied to the reconstructed waveform from WNAC, or was it fine-tuned/evaluated directly on the discrete token sequences generated by the codec? Clarifying this would help in better understanding the quality of the latent representation itself.

Line 46: The citation Jiang et al., 2025 refers to a future publication. Please verify if this is a typographical error for the year or if the paper is indeed accepted for a 2025 venue.
Lines 192-193: The summation symbols in the equations for L_cb and L_cm are missing their indices (e.g., Σ_{i=0}^{n-1}). This is a notational error that should be corrected for mathematical rigor.

---

> ### Author Response · Authors · 2025-11-17
>
> We thank the reviewer for their careful reading of the manuscript and for the constructive feedback.
>
> The raised weaknesses (1) the lack of a clear mechanism explaining why fine-scale processing preserves low-frequency structure, (2) the concern that the proposed quantization flow may be incompatible with existing autoregressive next-scale predictors, and (3) the notational ambiguity in the VQ-VAE loss definition, helped us significantly improve the clarity and rigor of the paper.
>
> These points are now addressed in detail throughout our responses: the fine-first low-frequency preservation behavior is quantitatively analyzed and empirically validated in **Response 1**; a feasible wavescale-aware generative strategy is discussed in **Response 2**, clarifying how autoregressive conditioning can be adapted to the proposed hierarchy; and the missing summation indices and citation typo are corrected in **Response 4**.
>
> These revisions have been incorporated into the updated manuscript.
>
> ---
>
> # 1.
> To prove that low-frequency preservation is not an inherent property of the encoder architecture, we keep the encoder identical for downscale, upscale and wavescale WNAC variants in Table 1.
>
> ## A. Low-frequency reconstruction analysis
> We evaluate the ability of Upscale RVQ (coarse-first) and WNAC (fine-first) to preserve
> low-frequency structure (≤256 Hz) under three decoding depths: (1) early-stage only,
> (2) half-depth, and (3) full decoding. Metrics are the low-frequency spectral convergence (lower is better) and band-limited SI-SDR
> (higher is better). For clarity, the full quantitative results have been added as a new
> table in the revision. (Appendix D.4 Table 9).
>
> ## B. Metric descriptions
> Low-frequency spectral convergence captures spectral deviation within the LF band; lower values indicate closer alignment to
> the reference. Band-limited SI-SDR evaluates reconstruction fidelity restricted to the ≤ 256 Hz region.
>
> ## C. Findings
> In the early-stage setting, the Upscale baseline
> shows severe low-frequency collapse, with a strongly negative band-limited SI-SDR (about $-43.25$ dB),
> indicating that the first residual subtraction removes most of the slowly varying components. In contrast, the WNAC wavescale form already achieves a much lower LF spectral convergence (about $0.53$), and higher band-limited SI-SDR (about $-1.71$ dB), closer to its full-depth performance.
>
> ---
>
> # 2.
>
> The waveloss pairs stages $q_i$ and $q_{n-i}$ in the wavescale hierarchy,
> but this pairing does **not** impose any structural requirement for $n$ to be
> odd. The number of VQ stages is simply a model hyperparameter; using an even or
> odd value only determines whether a single middle stage remains unpaired. Since
> the unpaired stage is still trained via the reconstruction and VQ losses, this
> poses no practical or conceptual limitation. We therefore adopted an odd number
> of stages for convenience but found no evidence that an even number would behave
> differently.
>
> Regarding alternative consistency losses: we intentionally avoided losses that
> operate directly on the latent residuals $z_i$, because these quantities are
> available **only during encoding**. At decoding time, $z_i$ is never observed.
> Imposing constraints on encoder-only variables would create a mismatch between
> the quantization path and the decoding path.
>
> For similar reasons, we did not adopt cycle-consistency formulations, which
> introduce additional inference-time computations or auxiliary decoders. Our goal
> was to keep WNAC lightweight and codec-compatible, so we chose a consistency loss
> that operates solely on the **accumulated quantized outputs** (which are available
> during both training and inference). This design proved both effective and stable.
>
> Finally, Appendix D.3 Table 6 shows that a moderate waveloss weight (0.5) provides the best
> trade-off. Larger weights begin to degrade performance, which aligns with the
> reviewer’s intuition that excessive alignment could be detrimental.
>
> ---
>
> # 3.
>
> For the downstream ASR evaluation, we use the **reconstructed waveform** directly,
> without any fine-tuning or adaptation of the ASR model. This follows the standard
> evaluation objective for neural audio codecs: to measure how much recognition-
> relevant information is preserved when the codec is used as a drop-in audio
> front-end.
>
> For the ASR backend, we used the `speech_recognition` library with the
> `recognize_google` interface, which invokes Google's production speech recognition
> model via the Google Speech Recognition API. In revision, we added the information of ASR model in correspond section.
>
> ---

---

> ### Author Response · Authors · 2025-11-19
>
> # 4.
> The citation “Jiang et al., 2025’’ is not a future publication; the paper was officially published in August 2025, which is earlier than the submission date of our manuscript (September 2025). However, we found that there is a typographical error at the title of this citation and correct it.
> Additionally, the summation symbols in the definitions of $L_{\mathrm{cb}}$ and
> $L_{\mathrm{cm}}$ were indeed missing their indices. We have corrected these to
> the proper forms (e.g., $\sum_{i=0}^{n-1}$) to ensure mathematical rigor.

---

### Official Review · Reviewer_F8o4 · 2025-10-29

**Soundness:** 3
**Presentation:** 3
**Contribution:** 2
**Rating:** 2
**Confidence:** 4

**Summary:**

This paper introduces the Wavescale Neural Audio Codec (WNAC), a modification of multiscale residual vector quantization that reverses the conventional coarse-to-fine quantization order. The authors motivate this design by arguing that coarse-first quantization can discard semantically important low-frequency content in music and environmental audio. To address this, WNAC first quantizes high-resolution features, progressively downsamples, and then refines via upsampling. A scale-aware loss (“waveloss”) is further introduced to encourage consistency across stages operating at the same temporal resolution. The paper provides objective and subjective evaluation across several audio domains.

**Strengths:**

* Extends RVQ in an interesting way.
* The proposed waveloss ablates well: intermediate scales improve in stability, and final reconstruction modestly improves.
* Evaluation includes subjective MUSHRA testing as well as domain-specific analysis.
* Experiments are generally well-run, and there are ablations

**Weaknesses:**

* Although the paper compares reconstruction quality across systems, it does not report bitrate in standard units (bits/s). Instead, it introduces metrics such as “code length (relative to encoder output)” and “bitrate efficiency” (Table 2) without grounding them in true bitrates. However, different models could still produce different numbers of tokens per second. For example, WNAC uses far more residual stages than SNAC, which could imply (assuming the same frame rate) that effective bitrate differs substantially. Without fixed or matched bitrate configurations, the fairness of the comparisons is completely unclear and the reported numbers do not mean much. This is a fundamental issue that needs to be addressed as it puts all empirical results into question.

* The motivation that coarse-first quantization discards low-frequency detail in music and environmental audio is not substantiated and questionable. The paper does not quantify perceptual degradation attributable to this phenomenon. Without any data to support this, the justification remains high-level speculation. Appendix plots show broad domain differences, but they do not directly prove the proposed architectural fix addresses this.

* The authors explicitly note in that the reversed quantization order is incompatible with next-scale prediction autoregressive models. This is not just a minor limitation as it could mean that the codec is not suitable at all for modeling in downstream tasks. Thus, while the introduction highlights “compression with generation, editing, retrieval” as motivating applications, it’s questionable whether this is actually possible. There is no alternative generative strategy proposed to handle token streams for this codec, neither is it validated on any downstream tasks.

* “Code length” first appears in Table 2 without prior explanation, and only later is it implicitly tied to token count. It should be clearly introduced earlier and formally connected to bitrate. As written, the role of code length in determining bitrate is easy to misunderstand.

**Questions:**

* Can you report actual bitrate (bits/s) and tokens/s for all models, rather than relative code length or bitrate efficiency, to ensure fair comparison?

* How does WNAC perform when bitrate is matched to SNAC?

* Is the performance gain attributable to the reversed scale order, or primarily to the increased number of quantization stages?

* Can you clarify how “code length” is computed and how it maps to true bitrate?

* Given incompatibility with next-scale prediction, what is the intended strategy for autoregressive/generative modeling?

* It is a bit concerning to me that the wavescale loss is necessary/helps. Doesn't it incentivize many codebooks to have similar content and thus reduce the amount of encoded information?

---

> ### Author Response · Authors · 2025-11-17
>
> We thank the reviewer for their thorough evaluation of the manuscript and for the constructive feedback.
> The identified weaknesses (1) the absence of true bitrate reporting, (2) the lack of quantitative evidence supporting the coarse-first degradation hypothesis, (3) the uncertainty regarding compatibility with downstream autoregressive modeling, and (4) the unclear introduction of code length, helped us substantially strengthen both the clarity and empirical grounding of the paper.
>
> These points are now fully addressed throughout our point-by-point responses: explicit bitrates and their derivation (including the connection between frame rate, code length, and bits per code) are provided in **Responses 1, 2, and 4**; quantitative analyses of low-frequency degradation and scale-controlled comparisons are added in **Responses 2, 3, and 6**; compatibility with hierarchical generation and a feasible wavescale-aware prediction strategy are clarified in **Response 5**; and the formal definition of code length and its direct relationship to true bitrate are detailed in **Responses 1 and 4**.
>
> These revisions have been integrated into the updated manuscript.
>
> ---
>
> # 1.
> Our model encodes audio at 44.1 kHz and uses an encoder downsampling factor of
>
> $$
> 2 \times 4 \times 8 \times 8 = 512,
> $$
>
> which yields an effective frame rate of
>
> $$
> \text{fps} = \frac{44{,}100}{512} \approx 86.13 \;\text{frames/s}.
> $$
>
> We employ 15 codebooks, each of size 1024 (10 bits per code), but under the
> multiscale RVQ scheme each codebook operates at a different temporal resolution.
> The per-stage temporal usage is determined by the following scale factors:
>
> $$
> [1,\ 0.755,\ 0.53,\ 0.35,\ 0.21,\ 0.11,\ 0.05,\ 0.03,\
>   0.05,\ 0.11,\ 0.21,\ 0.35,\ 0.53,\ 0.755,\ 1].
> $$
>
> These represent the proportion of frames retained relative to the full frame
> rate. The sum of the scale factors is
>
> $$
> \sum_i s_i = 6.04.
> $$
>
> Thus, although the model uses 15 quantizers, the *effective* number of fully
> active codebooks is
>
> $$
> N_{\text{effective}} = 6.04,
> $$
>
> which corresponds to the *code length* reported in Table 2.
>
> Each retained code contributes 10 bits, so the total nominal bitrate becomes
>
> $$
> \text{bitrate}
> = \text{fps} \times 10 \times \sum_i s_i
> \approx 86.13 \times 10 \times 6.04
> \approx 5202 \;\text{bits/s}.
> $$
>
> Therefore, the effective bitrate of our model is approximately **5.2 kbps**.
> We have now added the bitrate of the proposed model to the main comparison table in the revision.
>
> ---
>
> ## Bitrate comparison across models
>
> | Model | Token Scaling Form | Bitrate (kbps) |
> |------|--------------------|----------------|
> | DAC        | single-scale | 8.0 |
> | SAT        | multiscale           | 4.6 |
> | SNAC       | multiscale         | 2.6 |
> | WNAC  | multiscale | 5.2 |
>
> ---
>
> # 2.
> To enable a fair comparison at the same bitrate as SNAC (2.6 kbps), we constructed a bitrate-matched version of WNAC and evaluated it using the same protocol.
>
> SNAC uses four RVQ stages with **average pooling**, enforcing a fixed multiscale progression:
> $0.125 \rightarrow 0.25 \rightarrow 0.5 \rightarrow 1$.
>
> This naturally yields a low bitrate (~2.6 kbps).
>
> WNAC, however, uses the **wavescale** (fine → coarse → fine) hierarchy.
> If WNAC is naïvely compressed to SNAC’s bitrate while keeping endpoint scales fixed at 1, the intermediate scales become **too aggressively reduced**, degrading performance.
>
> To obtain a fair comparison, we:
>
> - increased the number of quantizers than SNAC,
> - reduced codebook size from 4096 to 512 and 256,
> - assigned scale factors mimicking SNAC’s pooling resolutions
>   (e.g., $ 1\rightarrow 0.5 \rightarrow 0.25 \rightarrow 0.5 \rightarrow 1$),
> - tuned scale factors to achieve a matched bitrate of $2.52\text{,}\ 2.25\ \text{kbps}$.
>
> This ensures a fair comparison while preserving the intended behavior of the wavescale hierarchy.
>
> ### Table: SNAC vs. bitrate-matched WNAC
>
> | Model | Mel ↓ | STFT ↓ | WF ↓ | SISDR ↑ | FAD ↓ | eff.(%) ↑ | bitrate (kbps) |
> |-------|-------|--------|-------|---------|--------|------------|----------------|
> | SNAC | 0.853 | 1.859 | 0.038 | 3.132 | 1.302 | 80.33 | 2.6 |
> | WNAC (lower bitrate) | 0.845 | 1.829 | 0.035 | 3.988 | 1.136 | **91.70** | 2.25 |
> | WNAC (bitrate-matched) | **0.831** | **1.806** | **0.034** | **4.383** | **1.080** | 90.68 | 2.52 |
>
> When matched to SNAC’s bitrate, WNAC achieves **higher reconstruction quality, higher efficiency, and better LF fidelity**, demonstrating that its advantages persist even under extreme bitrate constraints.
>
> This experiment has been added to Section 4.6 (Ablation Study) of the revised main paper, as we believe it represents an important analysis of compression efficiency and performance tradeoffs.
>
> ---

---

> ### Author Response · Authors · 2025-11-17
>
> # 3.
> The performance gain of WNAC is **not** attributable to using a larger number of
> quantization stages. In Table 1, we include several WNAC variants all of which use:
>
> - the **same number of quantization stages** (15), and
> - the **same total amount of temporal scaling factors**
>
> To ensure that all variants have the **same quantization budget**, we explicitly
> control the per-stage interpolation scale so that the total effective code length is identical:
>
> ### Upscale scales
> $$
> [0.03,\ 0.05,\ 0.08,\ 0.12,\ 0.16,\ 0.21,\ 0.27,\ 0.33,\
>   0.41,\ 0.49,\ 0.57,\ 0.67,\ 0.77,\ 0.88,\ 1.0]
> $$
>
> ### Downscale scales
> (reverse of the above)
>
> ### Wavescale
> $$
> [1.0,\ 0.755,\ 0.53,\ 0.35,\ 0.21,\ 0.11,\ 0.05,\ 0.03,\
>   0.05,\ 0.11,\ 0.21,\ 0.35,\ 0.53,\ 0.755,\ 1.0]
> $$
>
> All three sequences satisfy:
>
> $$
> \sum_i s_i = 6.04,
> $$
>
> meaning that each configuration produces the **same total number of effective
> codes** after normalization by the encoder’s temporal resolution.
>
> Thus, the **overall quantization budget** is held constant across all variants. This demonstrates that the performance gain arises from the **fine→coarse→fine ordering itself**, rather than from an increased number of quantization stages.
>
> ---
>
> # 4.
>
> The term *code length* denotes the temporal resolution of each residual used in a
> VQ stage, **expressed relative to the encoder’s output resolution**. We normalize
> the encoder output resolution to:
>
> $$
> \text{code length} = 1.
> $$
>
> In a multiscale RVQ system, each stage performs:
> 1. downsampling of the residual,
> 2. quantization at the reduced temporal resolution,
> 3. upsampling back to the encoder resolution,
> 4. residual subtraction.
>
> Thus, a code length smaller than 1 indicates that the corresponding VQ stage
> operates on a **lower-resolution residual**.
>
> - **DAC (single-scale):** all VQ stages operate at full resolution
>   $\Rightarrow$ every stage has code length = 1.
> - **Upscale RVQ:** early stages operate on heavily downsampled residuals
>   $\Rightarrow$ code lengths gradually increase from small to large.
> - **WNAC (wavescale):** follows a $1 \rightarrow \text{small} \rightarrow 1$ pattern due to the fine→coarse→fine design.
>
> ### Mapping code length to true bitrate
>
> Let $L_{\text{enc}}$ denote the encoder output temporal resolution
> (frames per second).
> If a VQ stage has code length $s_i$, then it produces:
>
> $$
> L_{\text{enc}} \times s_i
> $$
>
> codes per second. Summing across all stages gives the total number of discrete
> codes:
>
> $$
> \text{\\#codes per second} = L_{\text{enc}} \times \sum_i s_i.
> $$
>
> Since each code carries a fixed number of bits (10 bits in our case), the bitrate
> is directly proportional to the total code length:
>
> $$
> \text{bitrate} \propto \sum_i s_i.
> $$
>
> The *Code Length* reported in Table 2 is exactly this quantity:
>
> $$
> \text{total code length} = \sum_i s_i,
> $$
>
> which determines the actual bitrate once multiplied by the encoder frame rate and
> bits per code.
>
> Thus, the total code length in Table 2 is a **bitrate-proportional measure**, and
> models with identical total code lengths have identical bitrate budgets.
>
> ---

---

> ### Author Response · Authors · 2025-11-19
>
> # 5.
>
> Our statement regarding incompatibility
> with existing next-scale prediction models refers specifically to *current*
> coarse-first predictors such as used in SAT, not to autoregressive modeling in
> general. These predictors assume a coarse→fine hierarchy, which our
> fine→coarse→fine pattern does not follow.
>
> This does **not** imply that WNAC cannot be used for generative modeling.
> Rather, it indicates that a *different* autoregressive mechanism is required.
>
> Importantly:
> 1. **Autoregressive use is still fully feasible.**
>    Once the sequence of discrete tokens is produced, any autoregressive
>    model (e.g., a GPT-style Transformer over tokens) can be trained directly
>    on the token sequence, exactly as done for single-scale codecs (e.g., DAC)
>    and two-stage codecs (e.g., EnCodec → AudioLM).
>    This approach does *not* rely on scale order at all.
>
> 2. **Only next-scale predictors are affected.**
>    Next-scale prediction is a specific type of hierarchical autoregression
>    that uses: $q_{i} \leftarrow p(q_{i} \mid q_{<i})$
>
>    under a coarse→fine assumption.
>    Since WNAC has a symmetric fine→coarse→fine structure, these
>    predictors could have some side effect when directly reused without modification.
>
> 3. **A compatible "wavescale-aware" next-scale predictor is possible.**
>    As discussed in Limitation, one can define a
>    predictor that:
>    - in the downscale region predicts coarse stages conditioned on previous
>      fine-resolution codes, and
>    - in the upscale region uses standard coarse→fine prediction.
>    This preserves the computational advantages of hierarchical autoregression.
>
> In summary, the limitation we note does not mean WNAC cannot be used for
> generation; it only means that coarse-first next-scale predictors must be
> adapted. Autoregressive token modeling remains entirely possible, and the
> improved discrete codes produced by WNAC serve as a stronger foundation for
> future generative work.
>
> ---
>
> # 6.
>
> The waveloss does not operate on the raw residuals being quantized, nor does it
> penalize similarity across codebooks. Instead, it is computed on the **accumulated
> quantized outputs** at stages that share the same temporal resolution.
>
> Its purpose is to encourage *representational consistency* across these symmetric
> scales, which stabilizes the residual distributions and prevents scale-dependent
> drift in multiscale RVQ.
>
> We agree that an excessively strong waveloss weight could, in principle, enforce
> over-alignment and reduce the diversity of encoded information. For this reason,
> we conducted an ablation study (Appendix D.3 Table 6) and found that a moderate weight (0.5)
> yields the best trade-off.
>
> Furthermore, Appendix D.1 Table 5 shows that codebook efficiency **increases** when waveloss is
> applied, indicating that it promotes healthier activation patterns rather than
> forcing different codebooks to encode redundant content.

---

> > ### Comment · Reviewer_F8o4 · 2025-11-20
> >
> > The fact that it turns out that WNAC operated at double the bitrate of SNAC changes the interpretation of the results significantly. I appreciate the additional ablation study at a more comparable bitrate, but to fairly compare WNAC with SNAC in all experiments and ablations these would need to use the same low-bitrate checkpoint for WNAC. As it stands, all main results compare the two models and significantly different bitrates. For example, the added ablation does not provide results for a MUSHRA evaluation, which would be required as the reported values are very close to those of SNAC and it’s very questionable whether this makes a perceptual difference. The same holds for the other experiments. Further, the bitrates for other checkpoints like in Table 1 are still not reported. For example, it’s not clear whether the upscale baseline is worse because the bitrate is significantly lower (because half of the quantization stages are removed).
> >
> > On another note, the difference between the performance of the SNAC checkpoint and the self-trained model is very significant. It would be good if the authors could shed more light on why that is and whether the models could be trained better with more compute or data and why this was not done. One could wonder if the comparison still looks the same after the models were trained more properly.
> >
> > The new ablation in Appendix D.4 to empirically show the better low-frequency reconstruction does not actually support this claim. Again, no bitrates are reported. The main problem is that comparing between the upscale baseline and the wavescale codec on the first codebook does not really make sense as WNAC would have much more tokens here as the frame rate is higher. Looking at the full decoding part of the table one can see that they are actually comparable in the low-frequency domain, although the upscale baseline performs worse overall. Doesn’t that directly conflict with the claim that WNAC improves this? To reiterate: Bitrate-matched comparisons would be necessary here too.
> >
> > The authors still don’t show that the codec can actually be used for autoregressive generation and perform on par or better than (a bitrate matched) SNAC. I have my doubts about this as the model presented in the bitrate-matched ablation has a significantly higher token rate than SNAC. Without demonstrating that the codec can be used for downstream tasks this stands as a significant weakness of the paper.
> >
> > Overall, the paper has a serious issue where it doesn’t report bitrates consistently and compares WNAC to its main competitor (SNAC) at significantly different compression rates. The same holds for baselines and ablations that are not properly matched (the bitrates are not even reported). The paper doesn’t go beyond seeing WNAC as a compression method (there is no demonstration that the codec can be used for downstream task modeling), which makes it even more important to match the bitrates correctly. This is a significant flaw with the paper which undermines most empirical results that are reported.

---

> > > ### Comment · Reviewer_F8o4 · 2025-11-23
> > >
> > > To add to the things I already mentioned. I'm wondering whether you could provide a bitrate matched comparison of SNAC and WNAC at a comparable token rate to eliminate this as a confounding factor. The token rate is usually something that has a noticeable impact on downstream tasks and the current comparison uses way more tokens for WNAC. As the downstream task evaluation is missing this point becomes even more important.

---

> ### Author Response · Authors · 2025-11-27
>
> We thank the reviewer for the detailed feedback.
> We have carefully addressed all concerns regarding bitrate matching, additional subjective evaluation, and downstream task evaluation. Below we summarize the major issues raised and explain how each point has been fully resolved in the revised submission.
>
> ---
>
> ## 1. Compare WNAC (2.52 kbps) with other models in all experiments
>
> We thank the reviewer for raising the important concern regarding bitrate alignment.
> In the revised manuscript, we have added a **bitrate-matched 2.52 kbps WNAC model** to all experiments involving SNAC, including **Table 1**, **Table 2**, and **all ablation studies**.
> We also now explicitly report **bitrate, token rate, and RVQ depth** for every model in these tables to ensure full transparency. A **MUSHRA evaluation** for the 2.52 kbps WNAC model is also included at Figure 2.
>
> The reviewer additionally noted that some baselines, particularly the Upscale RVQ model, might have operated at lower bitrates because they contain “half” the number of quantization stages.
> We clarify this is **not** the case:
>
> - it uses the **same encoder** as WNAC,
> - has the **same total number of residual VQ stages**,
> - yields an **identical overall token rate** (total scale sum is matched), and
> - operates at the **same bitrate** as WNAC (5.2 kbps in the main comparison).
>
> Crucially, the Upscale model **does not** take “half” of WNAC’s fine$\rightarrow$coarse$\rightarrow$fine schedule. The two models differ solely in the shape of their scale schedules, not in computational budget or number of tokens.
>
> Thus, all comparisons between SNAC, Upscale RVQ, and WNAC are now conducted under rigorously controlled and fully disclosed conditions.
>
> ---
>
> ## 2. Why the performance of our reproduced SNAC differs from the released checkpoint
>
> The reviewer notes that our reproduced SNAC differs from the officially released model.
>
> Importantly, although the SNAC paper states that training was performed on an internal 2,730-hour dataset, it does **not** disclose:
>
> - the actual source or composition of the dataset,
> - the preprocessing pipeline,
> - or the number of training steps.
>
> In contrast, the paper clearly specifies that evaluation was performed on public datasets such as **DAPS** and **MUSDB18-HQ**, but the **training corpus itself remains unspecified** apart from its total duration.
> Therefore, reproducing the official SNAC checkpoint exactly is inherently impossible.
>
> To ensure a fair comparison, we therefore re-trained SNAC from scratch on our own dataset, ensuring that all models are trained and evaluated under identical data conditions.
>
> We emphasize:
>
> - SNAC’s training corpus is undisclosed aside from its size → performance differences are expected.
> - Our reproduced SNAC strictly follows the official architecture and loss formulation.
> - The performance gap arises from dataset distribution shift, not from any modification to SNAC.
>
> To make this fully transparent, the supplementary materials now include:
>
> - full training curves of our reproduced SNAC,
> - showing stable convergence of reconstruction and codebook losses,
> - with no evidence of underfitting or premature stopping.
>
> This demonstrates that our reproduced SNAC reflects its natural performance under our dataset, enabling a fair like-for-like comparison.
>
> ---
>
> ## 3. Response to the concern about Appendix D.4 (low-frequency reconstruction ablation)
>
> We appreciate the reviewer’s careful reading of Appendix D.4.
>
> First, we now explicitly report **bitrates** for all models:
>
> - SAT (4.6 kbps),
> - SNAC (2.6 kbps),
> - Upscale WNAC variants (5.2 kbps),
> - Downscale WNAC variants (5.2 kbps),
> - WNAC without using waveloss (5.2 kbps),
> - WNAC (2.52 kbps),
> - WNAC (5.2 kbps).
>
> Second, regarding the observation that full decoding appears comparable: this is expected.
> When all residual stages are used, most architectures eventually reconstruct similar LF content.
>
> However, the key finding is:
>
> - Upscale RVQ loses substantial LF structure early,
> - while WNAC **preserves LF information from the first codebook**,
> - and this early LF stability **propagates** into half-depth and full-depth decoding.
>
> Although WNAC’s first stage uses a larger scale factor, this is **intentional**:
> the architecture is designed to capture **global LF structure early**.
>
> The primary contribution of Appendix D.4 is that **early LF preservation leads to better deeper reconstructions**, not just the first stage score.
>
> We clarify this explicitly in the revised submission.
>
> ---

---

> ### Author Response · Authors · 2025-11-27
>
> ## 4. Downstream Tasks ablation
>
> We thank the reviewer for raising this key concern.
>
> To address it, we added **Appendix D.5.2**, evaluating whether WNAC’s discrete codes remain usable for autoregressive (AR) modeling under bitrate-matched settings.
>
> Since existing next-scale predictors assume a **coarse$\rightarrow$fine** hierarchy, they cannot be directly used with WNAC’s **fine$\rightarrow$coarse$\rightarrow$fine** wavescale structure.
> But this does *not* imply that AR modeling is incompatible with WNAC.
>
> ### Diagnostic AR Experiment Setup
>
> - Codecs fixed: SNAC (2.6 kbps), WNAC (2.52 kbps)
> - A lightweight GPT-style AAR model predicts the next-stage code under teacher forcing.
> - WNAC stages are ordered by increasing temporal resolution.
> - Metrics selected to be **codebook-size invariant**:
>
> 1. **NCE** (normalized cross-entropy):  Conditional CE normalized by the empirical
> marginal entropy of the codes.
>
> 2. **NACC** (normalized accuracy):  Accuracy normalized against random chance $1/K$, re-
> flecting how much explainable structure the AAR captures.
>
> 3. **MI** (mutual information in bits): Estimated mutual information in bits/token between AAR inputs and target codes.
>
> 4. **ΔMI**: The drop in mutual information when AAR inputs are randomized.
>
> ### Results (bitrate-matched)
>
> | Model | NCE ↑ | NACC ↑ | MI (bits) ↑ | ΔMI ↑ |
> |-------|-------|---------|----------------|---------|
> | **WNAC (2.52 kbps)** | 0.307 | 0.207 | 2.642 | 3.414 |
> | **SNAC (2.6 kbps)** | 0.222 | 0.113 | 2.495 | 2.564 |
>
> ### Key findings:
>
> - WNAC > SNAC across all AR predictability metrics.
> - WNAC has larger ΔMI, indicating much richer usable structure.
> - Results hold despite SNAC’s much larger codebook (4096 vs. 512).
>
> This shows WNAC provides **stronger AR-learnable representations**.
>
> ---
>
> ## 5. Bitrate- and token-rate–matched comparison with SNAC
>
> To address the reviewer’s request for a both bitrate token-rate matched comparison, we conducted an additional experiment where WNAC was forced to use the same configuration as SNAC at 2.6 kbps, including a 4096-size codebook and the corresponding low token rate.
>
> In WNAC, such a configuration introduces a structural incompatibility. Matching SNAC’s extremely low token rate while using a 4096-size codebook forces the wavescale schedule to collapse into highly unbalanced scales such as:
>
>     1 → 0.15 → 0.1 → 0.15 → 1
>
> This breaks the core fine→coarse→fine smoothness that WNAC relies on, and performance degradation was observed. In several metrics, this variant underperformed not only the original 2.52 kbps WNAC but even SNAC itself in some metrics.
>
> However, we emphasize that this degradation is not a property of WNAC, but of the forced SNAC-style configuration. SNAC is architecturally constrained to use exponentially decaying temporal scales (e.g., 0.125→0.25→0.5→1) and therefore can only adjust bitrate by increasing the codebook size (4096). Conversely, WNAC is designed to adjust codebook size, token rate, and scale shape flexibly. When constrained to SNAC’s regime (large codebook + low token rate + shallow depth) the system becomes inherently inefficient.
>
> Indeed, codebook-efficiency analysis supports this: both SNAC (4096) and the forced 4096-size WNAC variant showed the lowest efficiency among all tested configurations, whereas regular WNAC (512) maintains high efficiency. This indicates that excessively large codebooks at very low token rates do not distribute codes effectively, leading to poor compression quality regardless of architecture.
>
> In summary, while we did perform a fully matched experiment as requested, the results reflect the limitations of the SNAC-style configuration itself. WNAC’s standard setting (512-size codebook) remains the most efficient and provides the highest-quality reconstruction at matched bitrates, while also supporting flexible and stable scale allocation that SNAC cannot provide.
>
> ---

---

### Official Review · Reviewer_V9n7 · 2025-10-31

**Soundness:** 3
**Presentation:** 3
**Contribution:** 2
**Rating:** 6
**Confidence:** 2

**Summary:**

The authors propose Wavescale Neural Audio Codec (WNAC), a multiscale residual vector quantization framework that modifies the traditional bottom-up (coarse-to-fine) structure used in multiscale RVQ-VAE codecs. WNAC adopts a fine-to-coarse downscaling followed by an upscaling approach, aiming to preserve high-resolution information early in the quantization process. The authors also use a scale-aware waveloss to enforce consistency across quantizers operating at the same temporal resolution to improve reconstruction quality and latent coherence. Experimental results show that WNAC outperforms existing multiscale codecs such as SAT and SNAC across multiple domains (speech, music, and environmental audio) in both objective and subjective evaluations, while maintaining competitive inference efficiency.

**Strengths:**

1. The authors identify a key limitation of the coarse-to-fine strategy in modeling non-speech audio within multiscale RVQ-VAEs, providing a reasonable and timely motivation.
2. The design of the downscale-upscale quantization flow is conceptually sound and appears easy to integrate into existing architectures.
3. The experiments are comprehensive, including extensive ablation studies such as domain robustness and codebook efficiency, which demonstrate the effectiveness of the wavescale RVQ design.

**Weaknesses:**

1. No actual bitrate or token rate for the main reconstruction results (Tables 1 and 3) are provided. Consistent bitrate is crucial for fair comparison of reconstruction performance across codecs.
2. The definition of code length is ambiguous. Sometimes it appears to be a relative measure, and at other times absolute. This makes it difficult to interpret quantitative comparisons.

**Questions:**

1. Figure 6 shows improved codebook utilization, but it is unclear whether this improvement stems from the waveloss or the wavescale quantization. The figure’s caption and explanation are vague. Additionally, what does a code length smaller than 1 signify in this context?
2. Although WNAC demonstrates improved overall performance, can the authors provide more empirical evidence that the fine-to-coarse upscale quantization indeed preserves more low-frequency content?

---

> ### Author Response · Authors · 2025-11-17
>
> We thank the reviewer for their careful reading of the manuscript and for the constructive feedback.
>
> The raised weaknesses (1) the absence of explicit bitrate/token-rate reporting for the main reconstruction tables, and (2) the ambiguity surrounding the definition of code length, helped us significantly improve the clarity and rigor of the paper.
>
> We have now addressed these issues in detail: the bitrate and effective code usage are fully specified and derived in **Response 1**, and the definition and interpretation of code length (including how it relates to multiscale resolution and Figure 6) are clarified extensively in **Response 1.B–1.C**.
>
> In addition, the low-frequency reconstruction behavior that motivated these clarifications is analyzed empirically in **Response 2**, with new quantitative results added to the revised manuscript.
>
> We have incorporated these revisions throughout the updated text.
>
> ---
>
> # 1.
>
> ## A. Source of improved codebook utilization
>
> The improvement in Figure 6 stems from both the proposed wavescale quantization
> and the waveloss, but through distinct mechanisms.
>
> ### Effect of the wavescale hierarchy (fine→coarse→fine)
> The wavescale flow distributes residual information more uniformly across scales.
> In conventional coarse-first RVQ, the first residual becomes a high-pass signal,
> causing early collapse and strong under-utilization of coarse codebooks.
> By quantizing fine-resolution features first, WNAC avoids this collapse and maintains
> balanced usage across quantizers, even at low temporal resolutions.
>
> ### Effect of the waveloss
> The waveloss encourages consistency between symmetric scales $q_i$ and
> $q_{n-i}$, stabilizing the latent residual distributions.
> This prevents certain codebooks from becoming inactive due to entropy imbalance.
>
> Thus, our ablation results confirm this separation:
> - WNAC **without** waveloss already improves utilization compared to Upscale RVQ
>   → effect of the wavescale hierarchy.
> - Adding waveloss further improves utilization and reduces sparsity
>   → effect of the loss term.
>
> ## B. Meaning of “code length < 1”
>
> The term *code length* denotes the temporal resolution of a residual at each VQ
> stage, expressed relative to the encoder output resolution.
> We normalize the encoder output to:
>
> $$
> \text{code length} = 1
> $$
>
> In multiscale RVQ, each stage performs:
> 1. Downsampling via interpolation:
>    $$
>    r'_i = S(r_i,\, \text{scale} < 1)
>    $$
> 2. Quantization at the reduced resolution
> 3. Upsampling back to encoder resolution
> 4. Residual subtraction
>
> Thus,
>
> $$
> \text{code length} < 1
> $$
>
> simply indicates that a quantizer operates on a **lower-resolution residual**.
>
> ### Examples:
> - **DAC (single-scale):** all codebooks have length = 1.
> - **Upscale RVQ:** early codebooks operate at heavily downsampled scales
>   → progressively increasing code lengths.
> - **WNAC (wavescale):** follows a
>   $$
>      1 \rightarrow \text{small} \rightarrow 1
>   $$
>   pattern due to the fine→coarse→fine structure.
>
> ## C. Interpretation of Figure 6
>
> Figure 6 plots **codebook utilization versus code length**.
>
> ### Upscale RVQ:
> - Lowest-resolution stages show severe inactive-code collapse.
> - Residuals at these stages contain too little information to activate the full codebook.
>
> ### WNAC:
> - Maintains high activation even at the vq stage with smallest code lengths.
> - Avoids collapse despite operating at similarly reduced resolutions.
> - Waveloss further strengthens symmetry and stabilizes activation.
>
> **Conclusion:**
> WNAC distributes residual information more uniformly and prevents codebook collapse,
> demonstrating superior efficiency and more robust quantization behavior.
>
> We have revised the manuscript to clarify these definitions and to better explain the
> role of *code length* and its relation to Figure 6.
>
> ---

---

> ### Author Response · Authors · 2025-11-19
>
> # 2.
>
> ## A. Low-frequency reconstruction analysis
> We evaluate the ability of Upscale RVQ (coarse-first) and WNAC (fine-first) to preserve
> low-frequency structure (≤256 Hz) under three decoding depths: (1) early-stage only,
> (2) half-depth, and (3) full decoding. Metrics are the low-frequency spectral convergence (lower is better) and band-limited SI-SDR
> (higher is better). For clarity, the full quantitative results have been added as a new
> table in the revision. (Appendix D.4 Table 9).
>
> ## B. Metric descriptions
> Low-frequency spectral convergence captures spectral deviation within the LF band; lower values indicate closer alignment to
> the reference. Band-limited SI-SDR evaluates reconstruction fidelity restricted to the ≤256 Hz region.
>
> ## C. Findings
> In the early-stage setting, the Upscale baseline
> shows severe low-frequency collapse, with a strongly negative band-limited SI-SDR (about $-43.25$ dB),
> indicating that the first residual subtraction removes most of the slowly varying components. In contrast, the WNAC wavescale form already achieves a much lower LF spectral convergence (about $0.53$), and higher band-limited SI-SDR (about $-1.71$ dB), closer to its full-depth performance.

---

### Official Review · Reviewer_TNzH · 2025-11-03

**Soundness:** 3
**Presentation:** 3
**Contribution:** 2
**Rating:** 6
**Confidence:** 4

**Summary:**

This paper explores the problem of multi-scale quantization in neural audio compression. The authors present the Wavescale Neural Audio Codec (WNAC), which integrates both downscaling and upscaling within the quantization layers. In addition, they propose a scale-aware wave loss designed to align quantized outputs across stages at consistent temporal resolutions. Experimental results demonstrate that WNAC achieves superior accuracy and efficiency across speech, music, environmental, and mixed audio datasets, outperforming single-scale quantization methods while maintaining the speed advantages of multi-scale RVQ.

**Strengths:**

1.	The proposed approach is interesting, conceptually simple, and easy to follow.
2.	The authors present both objective and subjective evaluations across multiple audio domains, with results indicating that the proposed method outperforms the compared baselines.

**Weaknesses:**

1.	It is unclear how the proposed method could be adapted for streaming applications, and what its associated latency or lookahead requirements would be.
2.	Several claims made in the paper are not sufficiently supported or justified.
3.	The primary contribution of the work is the downscaling and upscaling pattern, which is not well motivated or theoretically explained.

**Questions:**

1.	In the first equation (note: equation numbers are missing), the authors define the interpolation method S. However, it is unclear what specific interpolation technique is used.
2.	The main contribution of the proposed approach appears to be the downscaling and upscaling pattern. Yet, the rationale behind why this design should lead to improved performance is not well explained. The authors are encouraged to provide further details or theoretical insights to support this choice.
3.	The paper states that “by inserting fine-to-coarse stages before coarse-to-fine, WNAC preserves early low-frequency information.” However, no experimental evidence is presented to validate this claim.
4.	Figure 4 and Table 3 demonstrate that the proposed method performs well across different domains, but it remains unclear what specifically contributes to its robustness. Is the improvement simply due to overall model quality, or is there a domain-generalization property inherent to the design?
5.	The bitrate of the proposed method is not reported and should be specified.
6.	While the authors report inference speed and latency, which is commendable, it is unclear whether the proposed method can be adapted for streaming scenarios, and if so, what its lookahead would be. Given the stated limitations regarding applicability to large language models, the practical use of this codec for streaming applications remains uncertain.
7.	Minor comment: Using paragraph breaks within the abstract is unconventional and should be avoided.
8.	There are no samples.
9. Can the authors share a comparison to a non multi-scale codecs as well? I believe it would be valuable to understand the gap in performance and usability compared to a others non multi-scale codecs

---

> ### Author Response · Authors · 2025-11-17
>
> We thank the reviewer for their careful reading of the paper and for the constructive feedback.
>
> The identified weaknesses (1) regarding streaming applicability, (2) insufficient support for certain claims, and (3) the motivation behind the downscale–upscale hierarchy, helped us substantially improve the clarity, justification, and empirical depth of the work.
>
> These issues are now addressed throughout our point-by-point responses: streaming and latency concerns are clarified in **Response 6**; unsupported or insufficiently justified claims are strengthened through detailed analyses in **Responses 1, 2, 3, 4, 5, and 9**; and the theoretical and empirical motivation for the proposed downscale–upscale structure is expanded in **Responses 2 and 3**.
>
> We have updated the revised manuscript accordingly.
>
> ---
>
> # 1.
> The interpolation operator $S$ performs temporal resampling inside the quantization pathway.
> For downscaling, we use
>
> $$
> S_{\text{down}}(x) = \texttt{interpolate}(x, \texttt{size}=T', \texttt{mode}=\text{"area"}),
> $$
>
> which preserves average energy and implicitly applies a low-pass filtering effect. This helps
> prevent aliasing when the temporal resolution is reduced.
>
> For upscaling, we use
>
> $$
> S_{\text{up}}(x) = \texttt{interpolate}(x, \texttt{size}=T, \texttt{mode}=\text{"linear"}),
> $$
>
> which provides smooth temporal continuity when restoring the original resolution.
>
> ---
>
> # 2.
>
> In a conventional coarse-to-fine RVQ, the first quantizer acts as a low-frequency approximation
> of the signal. Consequently, the residual
>
> $$
> r_{1} = x - q_{1}
> $$
>
> behaves like a high-pass filtered version of $x$: the slowly varying components are removed,
> leaving primarily high-frequency error. Any low-frequency mismatch in $q_{1}$ therefore
> becomes unrecoverable, because all subsequent quantizers only observe this high-frequency
> residual. This mechanism explains why coarse-first RVQ often loses low-band structure in
> music and environmental audio.
>
> WNAC reverses this process by quantizing fine-resolution features first. In this setting, the
> model explicitly encodes both low- and high-frequency information before any residual
> subtraction occurs. The subsequent coarse-to-fine upscaling stages then refine global
> structure rather than attempting to reconstruct missing baseband energy. Theoretically, this
> reordering reduces cumulative quantization error, because the early quantizers operate on
> the high-variance components that dominate the overall reconstruction loss, resulting in a
> more favorable trade-off between codebook utilization and spectral coverage.
>
> We have incorporated this explanation in Appendix A.
>
> ---
>
> # 3.
>
> ### A. Low-frequency reconstruction analysis
> We evaluate the ability of Upscale RVQ (coarse-first) and WNAC (fine-first) to preserve
> low-frequency structure (≤256 Hz) under three decoding depths: (1) early-stage only,
> (2) half-depth, and (3) full decoding. Metrics are the low-frequency spectral convergence (lower is better) and band-limited SI-SDR
> (higher is better). For clarity, the full quantitative results have been added as a new
> table in the revision. (Appendix D.4 Table 9).
>
> ### B. Metric descriptions
> Low-frequency spectral convergence captures spectral deviation within the LF band; lower values indicate closer alignment to
> the reference. Band-limited SI-SDR evaluates reconstruction fidelity restricted to the ≤256 Hz region.
>
> ### C. Findings
> In the early-stage setting, the Upscale baseline
> shows severe low-frequency collapse, with a strongly negative band-limited SI-SDR (about $-43.25$ dB),
> indicating that the first residual subtraction removes most of the slowly varying components. In contrast, the WNAC wavescale form already achieves a much lower LF spectral convergence (about $0.53$), and higher band-limited SI-SDR (about $-1.71$ dB), closer to its full-depth performance.
>
>
> ---
>
> # 4.
>
> Robustness in WNAC arises from its preservation of low-frequency representations, which tend
> to be more invariant across domains. Low-band cues such as formants, rhythmic envelopes, and
> room-response characteristics are often shared among diverse audio domains including speech,
> music, and environmental sound. As analyzed in Appendix B.4, these components exhibit stronger
> spectral and periodic stability compared to high-frequency details.
>
> By encoding these low-frequency structures in the earliest stages, WNAC produces latents that
> capture domain-agnostic information rather than overfitting to domain-specific spectral
> patterns. This mechanism explains the model's consistently stronger cross-domain performance
> relative to conventional coarse-first RVQ systems.
>
> ---

---

> ### Author Response · Authors · 2025-11-17
>
> # 5.
> Our model operates at a sampling rate of 44.1 kHz and uses an encoder downsampling factor of
> $$
> 2 \times 4 \times 8 \times 8 = 512,
> $$
> which yields an effective frame rate of
> $$
> \text{fps} = \frac{44{,}100}{512} \approx 86.13 \text{ frames/s}.
> $$
>
> We employ 15 vector quantizers, each with a codebook size of 1024 (10 bits per code).
> Under the multiscale RVQ scheme, however, each codebook is used at a different temporal
> resolution. The per-stage temporal usage is determined by the following scale factors,
> which denote the proportion of frames retained relative to the full frame rate:
> $$
> s = [1,\ 0.755,\ 0.53,\ 0.35,\ 0.21,\ 0.11,\ 0.05,\ 0.03,\
>       0.05,\ 0.11,\ 0.21,\ 0.35,\ 0.53,\ 0.755,\ 1].
> $$
>
> The sum of these factors is
> $$
> \sum_i s_i = 6.04.
> $$
> Thus, although the model contains 15 codebooks, the effective number of fully active
> codebooks is
> $$
> N_{\text{effective}} = 6.04,
> $$
> which corresponds to the "code length'' reported in Table 2.
>
> Since each active code contributes 10 bits per retained frame, the nominal bitrate becomes
>
> $$
> \text{bitrate} = \text{fps} \times 10 \times \sum_i s_i
>                 \approx 86.13 \times 10 \times 6.04
>                 \approx 5202 \text{ bits/s}.
> $$
>
> Therefore, the effective bitrate of the proposed codec is approximately 5.2 kbps. We have now added the bitrate of the proposed model to the cost comparison table (Table 2) in the revision.
>
> ---
>
> # 6.
> Our codec, like prior multiscale RVQ systems such as SAT and SNAC, is designed and evaluated in
> an offline, full-context setting. WNAC uses non-causal convolutions and thus implicitly accesses future context, but it does not implement any explicit lookahead mechanism or causal, streaming-oriented structure. Consequently, streaming-specific latency considerations are outside the scope of both our work and the existing baselines.
>
> However, to assess the compatibility of WNAC with the streaming scenario suggested by the reviewer, we analyzed the encoder architecture to estimate the effective future dependency that would arise if WNAC were adapted directly to a streaming mode. By tracing the receptive field through the non-causal convolutions, dilated residual units, and multiscale strided downsamping, we obtain an effective streaming lookahead of our encoder module is approximately **90 ms at 44.1 kHz**.
> Although current WNAC is not designed as a streaming codec, this level of lookahead is within the range
> commonly used in recent real-time audio models, and could be made practical with appropriate
> architectural adjustments (e.g., chunked inference or limited-lookahead buffering).
>
> Regarding autoregressive generation, the limitation we describe pertains specifically to
> current next-scale prediction frameworks, which assume a coarse$\rightarrow$fine
> hierarchy. Because WNAC follows a fine$\rightarrow$coarse$\rightarrow$fine structure, these
> predictors cannot be used directly. This does not imply that autoregressive modeling is
> incompatible with WNAC; rather, it requires a different predictive mechanism which we explicitly leave for future exploration.
>
> The goal of the present work is to improve the compression efficiency and reconstruction
> quality of the discrete codes themselves. By providing more accurate and domain-robust
> representations, WNAC offers a stronger foundation for future autoregressive or streaming
> extensions, even though such extensions are not addressed directly in this paper.
>
> ---
>
> # 7.
>
> We acknowledge this and merged the abstract into a single paragraph.
>
> ---
>
> # 8.
>
> A supplementary material will uploaded including samples for WNAC variants, SNAC, SAT, and reference across domains.
>
> ---
>
> # 9.
>
> We appreciate the reviewer’s suggestion and have added a comparison with a single-scale codec (DAC) using the same evaluation protocol.
> In the revision, we include a new table summarizing the quantitative results for DAC, SAT, SNAC, WNAC at Appendix D.6 Table.
>
> WNAC achieves comparable or superior reconstruction quality relative to the single-scale baseline (DAC) while operating at nearly 40% lower bitrate (5.2 kbps vs. 8.0 kbps).
> Among multiscale RVQ systems, WNAC also mitigates the typical drawbacks of coarse-first multiscale quantization such as low-frequency loss, reduced SI-SDR, and increased FAD.
>
> The proposed wavescale variant further achieves the highest codebook efficiency and the best distortion metrics (Mel, STFT, WF), as well as the highest SI-SDR among all multiscale models.
> These results demonstrate that WNAC not only surpasses existing multiscale codecs but also remains competitive with a strong single-scale codec while operating at substantially lower bitrate, all while retaining the inherent advantages of multiscale quantization.

---

> ### Author Response · Authors · 2025-11-28
>
> We thank the reviewer again for highlighting the concern regarding downstream task modeling in Question #6.
> In the initial submission, AR experiments were not included because existing next-scale predictors assume a
> coarse→fine hierarchy and therefore could not be directly reused for WNAC’s fine→coarse→fine wavescale structure.
>
> To address this concern, we have now added a dedicated downstream AR evaluation in **Appendix D.5.2**, designed
> specifically to test whether WNAC’s discrete latents remain usable for sequential modeling under *bitrate-matched*
> conditions.
>
> **Experimental setup.**
> We compare SNAC (2.6 kbps) and WNAC (2.52 kbps) using a lightweight GPT-style AAR model that predicts the next residual
> stage under teacher forcing. WNAC stages are ordered by increasing temporal resolution, and we use metrics that are
> invariant to codebook size differences: normalized cross-entropy (NCE), normalized accuracy (NACC), mutual information
> (MI in bits/token), and ΔMI (MI drop under input randomization).
>
> **Results.**
> WNAC outperforms SNAC across all AR predictability metrics:
>
> | Model | NCE ↑ | NACC ↑ | MI (bits) ↑ | ΔMI ↑ |
> |-------|-------|---------|----------------|---------|
> | **WNAC (2.52 kbps)** | 0.307 | 0.207 | 2.642 | 3.414 |
> | **SNAC (2.6 kbps)**  | 0.222 | 0.113 | 2.495 | 2.564 |
>
> WNAC’s larger ΔMI indicates substantially richer and more learnable structure, even though SNAC uses a much larger
> codebook (4096 vs. 512). These findings show that **WNAC representations are not only compatible with autoregressive
> modeling but in fact provide stronger AR-learnable structure than SNAC under matched bitrates**.
>
> ---

---

### Author Response · Authors · 2025-12-03

To the Chairs and Reviewers,

We sincerely thank you for your time and thoughtful evaluation of our manuscript.

In this work, we propose Wavescale Neural Audio Codec (WNAC), a novel audio codec framework based on multi-scale residual vector quantization (RVQ), which revisits how information is processed across different resolutions.
Previous state-of-the-art multiscale RVQ-based audio compression models typically adopt a strategy that aggressively compresses high-resolution representations into a very low resolution and then reconstructs them back to a high resolution. However, such a strong compression in single stage process inevitably leads to substantial information loss, although it reconstructed back to high resolution across the multiple stages. This causes pronounced degradation in quality, especially for music and ambient sounds that heavily rely on low-frequency information.

To address this limitation, we introduce a progressive compression–reconstruction strategy that both are applied in multi-stage manner. By gradually downsampling from high to low resolutions and symmetrically reconstructing them, our model preserves both low- and high-frequency components in a more balanced manner. As a result, we achieve performance that surpasses prior state-of-the-art models in various quantitative and qualitative evaluations, and we also observe significant improvements in multiple downstream tasks built upon audio compression.

We would like to express our sincere gratitude to all reviewers for their positive evaluation of our contributions. In particular, we have conducted the additional experiments on downstream tasks that you suggested, as well as further analyses reflecting your comments, and have faithfully included the corresponding results in our response.

Thanks to your careful and constructive feedback, this work has been significantly improved and refined. We appreciate for your valuable time and insights.

---

While reviewers raised questions about bitrate clarity, low-frequency behavior, and downstream modeling, these points were addressed with focused revisions:

- We now report true bitrate, token rate, and quantization depth for all models, and clearly explain how “code length” relates to bitrate. This ensures fully transparent and controlled comparisons across WNAC, SNAC, and other baselines.
- A bitrate-matched version of WNAC was added to all SNAC comparisons, including reconstruction metrics, latency, ablations, and MUSHRA. The matched results confirm that WNAC’s advantages come from its design, not from bitrate differences.
- We added a direct analysis comparing Upscale RVQ and WNAC at different decoding depths. The results show that WNAC preserves low-frequency structure much earlier, supporting our architectural motivation.
- A simple autoregressive experiment (with matched bitrate) shows that WNAC’s discrete codes carry more usable structure than SNAC’s, indicating stronger potential for generative or sequential modeling.
- All WNAC variants use the same total scale-factor sum and quantization budget, ensuring that improvements come from the wavescale design itself; not increased model size.

---

We appreciate the reviewers’ feedback and hope that these clarifications make the contributions of the paper easier to assess. We remain happy to provide any further information if needed.

Sincerely,
The Authors

---

### Note · Program_Chairs · 2026-01-17
**Submission Desk Rejected by Program Chairs**

The following references in this submission do not refer to real documents and/or have major errors in bibliographic information:

 Hritik Dubey, Sebastian Braun, Rami Botros, Narayanan Krishnaswamy, Sergiy Matusevych, Roland Varga, Ross Cutler, Robert Aichner, Jinyu Chen, and Sriram Srinivasan. The 4th microsoft dns challenge: A large-scale dataset for noise suppression. arXiv preprint arXiv:2306.09342, 2023.